# Identification of a novel enhancer essential for *Satb1* expression in T$_H$2 cells and activated ILC2s

Aneela Nomura[1], Tetsuro Kobayashi[2], Wooseok Seo[1], Michiko Ohno-Oishi[1], Kiyokazu Kakugawa[3], Sawako Muroi[1], Hideyuki Yoshida[4], Takaho A Endo[5], Kazuyo Moro[2,6], Ichiro Taniuchi[1]

The genome organizer, special AT-rich binding protein-1 (SATB1), functions to globally regulate gene networks during primary T cell development and plays a pivotal role in lineage specification in CD4+ helper-, CD8+ cytotoxic-, and FOXP3+ regulatory-T cell subsets. However, it remains unclear how *Satb1* gene expression is controlled, particularly in effector T cell function. Here, by using a novel reporter mouse strain expressing SATB1-Venus and genome editing, we have identified a *cis*-regulatory enhancer, essential for maintaining *Satb1* expression specifically in T$_H$2 cells. This enhancer is occupied by STAT6 and interacts with *Satb1* promoters through chromatin looping in T$_H$2 cells. Reduction of *Satb1* expression, by the lack of this enhancer, resulted in elevated IL-5 expression in T$_H$2 cells. In addition, we found that *Satb1* is induced in activated group 2 innate lymphoid cells (ILC2s) through this enhancer. Collectively, these results provide novel insights into how *Satb1* expression is regulated in T$_H$2 cells and ILC2s during type 2 immune responses.

## Introduction

The nuclear protein, special AT-rich binding protein 1 (SATB1), functions as a genome organizer and regulates highly ordered chromatin structures by tethering specialized AT-rich genomic regions, such as base unpairing regions (1, 2). SATB1 epigenetically regulates gene expression by recruiting various chromatin modifiers and nucleosome remodelling and deacetylase (NURD) complexes (3) and promote heterochromatin formation. Multiple studies have delineated the role of SATB1 for postnatal neuronal development and function (4, 5). In addition, SATB1 is highly expressed in the thymus and is deemed essential for the development of mature thymocytes (6). CD4+CD8+ double-positive (DP) immature thymocytes undergo a TCR-mediated selection process, known as positive selection, to become mature thymocytes that are committed to become either helper-(T$_H$), cytotoxic-(T$_C$) or regulatory-T (Treg) cells. In this context, SATB1 is required to control expression of genes encoding lineage specification transcription factors, *Thpok, Runx3,* and *FoxP3* for T$_H$, T$_C$, and Treg cells, respectively. In the absence of SATB1, aberrant expression of these transcription factors occurs through their derepression. For instance, *Thpok* expression was induced in MHC-I-restricted T$_C$ cells and *Foxp3* expression was induced in conventional CD4 T cells (7, 8).

The functions of SATB1 extend into the differentiation of effector T cells after encountering antigens in the periphery. *Satb1* expression is thought to be under the control of TCR signalling (9), which is counteracted by TGF-$\beta$ signalling (10). In these effector T cells, SATB1 functions to repress PD-1 expression and suppress T cell exhaustion (10). Lastly, *Satb1* expression is increased upon IL-23 stimulation in pathogenic IL-17-producing T$_H$17s and promotes their pathogenicity in experimental autoimmune encephalomyelitis via regulation of GM-CSF production and suppression of PD-1 (11). The role of SATB1 in CD4+ type 2 helper (T$_H$2) differentiation has been characterized but has found very contradictory results. Based on the identification of SATB1-binding sites in the *T$_H$2* locus containing *Il-5, Il-4,* and *Il-13* genes in a murine T$_H$2 cell line in vitro, the first report claimed that SATB1 functioned as a positive regulator of T$_H$2 cytokine expression (12). Other studies, however, demonstrated that SATB1 only represses IL-5 expression in human CD4 T$_H$2 cultures (13). It was also reported that SATB1 cooperates with $\beta$-catenin to control the expression of *Gata3*, the key T$_H$2 lineage transcription factor critical for T$_H$2 differentiation and function (14). However, conditional loss of *Satb1* in CD4+ T cells, using the ThPOK-Cre mouse strain, showed that loss of *Satb1* expression had no detrimental effects on T$_H$2 differentiation, at least in vitro. Furthermore, it has been shown that the expression of *Satb1* is controlled by both IL-4 and NF$\kappa$B signalling in vitro (15).

[1]Laboratory for Transcriptional Regulation, RIKEN Center for Integrative Medical Sciences (IMS), Yokohama, Japan   [2]Laboratory for Innate Immune Systems, RIKEN Center for Integrative Medical Sciences (IMS), Yokohama, Japan   [3]Laboratory for Immune Crosstalk, RIKEN Center for Integrative Medical Sciences (IMS), Yokohama, Japan   [4]Laboratory for YCI Laboratory for Immunological Transcriptomics, RIKEN Center for Integrative Medical Sciences (IMS), Yokohama, Japan   [5]Laboratory for Integrative Genomics, RIKEN Center for Integrative Medical Sciences (IMS), Yokohama, Japan   [6]Laboratory for Innate Immune Systems, Department of Microbiology and Immunology, Graduate School for Medicine, Osaka University, Osaka, Japan

Correspondence: ichiro.taniuchi@riken.jp
Wooseok Seo's present address is Department of Immunology, Nagoya University Graduate School of Medicine, Nagoya, Japan
Michiko Ohno-Oishi's present address is Department of Ophthalmology, Tohoku University Graduate School of Medicine, Sendai, Japan

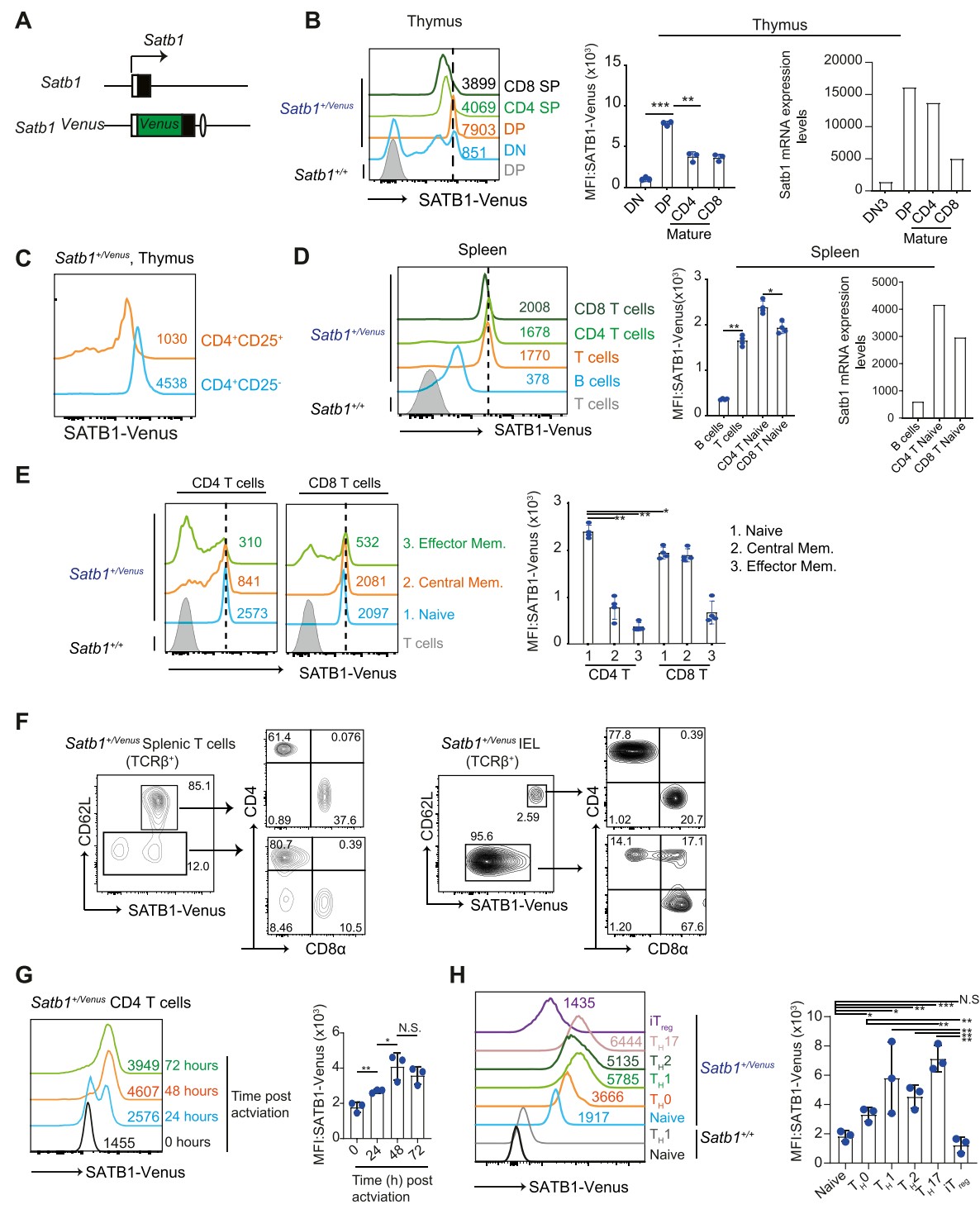

**Figure 1. Expression of SATB1-venus reporter during T cell development and activation.**

**(A)** Schematic shows *Satb1-Venus* gene locus in *Satb1^Venus* knockin mice. **(B)** Histograms, on the left, show expression of SATB1-Venus in CD4⁻CD8⁻ DN, CD4⁺CD8⁺ DP, CD24^loTCRβ^hiCD4⁺CD8⁻ mature SP and CD24^loTCRβ^hiCD4⁻CD8⁺ mature SP thymocytes of *Satb1^+/Venus* mice. Numerical values indicate SATB1-VENUS MFI (mean fluorescence intensity) and data are a representative of three biological experiments and summarised in the middle graph. Statistics were calculated by two-way ANNOVA Tukey`s multiple comparisons: *p- < 0.05, **p- < 0.01, ***p- < 0.001. Graph, on the right, shows *Satb1* mRNA expression in DN3, DP, CD4 SP and CD8 SP thymocytes, derived from IMMGEN RNA-seq database. **(C)** Histograms show SATB1-Venus expression in conventional CD4⁺CD25⁻ versus regulatory CD4⁺CD25⁺ mature thymocytes of *Satb1^+/Venus* mice. Numerical values indicate SATB1-Venus MFI. Data are representative of three biological experiments. **(D)** Histograms, on the left, show expression of SATB1-Venus in splenic B cells (B220⁺TCRβ⁻), T cells (B220⁻TCRβ⁺), CD4 T cells (TCRβ⁺CD4⁺CD8α⁻), and CD8 T cells (TCRβ⁺CD4⁻CD8α⁺) of *Satb1^+/Venus* mice. Data are summarised in the middle graph from three biological experiments and statistics were calculated by two-way ANNOVA Tukey`s multiple comparisons: *P- < 0.05, **P- < 0.01, ***P- < 0.001. Graph on the right shows *Satb1* expression in B cells, naïve CD4, and naïve CD8 T cells from IMMGEN RNA-seq database. **(E)** Histograms show SATB1-Venus expression in CD4

Currently, there is very little mechanistic insight to how *Satb1* gene expression is transcriptionally induced in T cells. In mice, there are four alternative promoters, P1, P2, P3, and P4 generate *Satb1-1a*, *Satb1-1b*, *Satb1-1c*, and *Satb1-1d* transcripts, respectively. Usage of these four *Satb1* promoters is differentially regulated in CD4 $T_H2$ in vitro (15, 16). Interestingly, a GWAS study mapped a Psoriasis-associated single nucleotide polymorphism (SNP, rs73178598) around 240 kb upstream of the *SATB1* locus, where an antisense noncoding RNA called *SATB1-AS1* is transcribed (17). This study hinted the presence of *cis*-regulatory genomic elements that could control *Satb1* expression in various effector T cells and contribute to tissue inflammation. Thus, it is quite important to define *Satb1* expression during effector T cell differentiation and how it is controlled.

In this study, we used a novel *Satb11-Venus* reporter strain and a genome editing technology to identify a novel IL-4-responsive $T_H2$-specific enhancer for *Satb1*, defined as *Satb1-Eth2*. *Satb1-Eth2* is essential to maintain *Satb1* expression, not only in CD4 $T_H2$ cells but also in activated ILC2s. Furthermore, loss of *Satb1-Eth2* resulted in elevated IL-5 expression in CD4 $T_H2$ cells. Collectively, our study unravels mechanisms by which *Satb1* expression is retained in immune cells mediating type 2 immune responses.

# Results

### Strict control of *Satb1* expression revealed by a *Satb1-Venus* fusion reporter

To quantify SATB1 protein expression in various types of murine cells, particularly in T cells and at single cell level, we generated a *Satb1^Venus* allele by the knock-in approach to insert the Venus open reading frame, downstream of the initiation codon (in exon2) of the *Satb1* gene (Figs 1A and S1A and B). In *Satb1^{+/Venus}* thymocytes, the *Satb1^Venus* allele expressed the SATB1-Venus protein (Fig S1C) that showed a cage-like distribution in the nucleus, as previously reported (Fig S1D) (18). Phenotypic analyses in *Satb1^{Venus/Venus}* mice did not show abnormalities in thymocyte development, that was observed in *Satb1^{Flx/Flx} Cd4-cre* mice, and the numbers of mature CD4 and CD8 SP thymocytes were comparable with the wildtype (Fig S1E). Thus, the SATB1-Venus fusion protein is likely to retain endogenous SATB1 function to support, at least, primary T cell development. Using *Satb1^{+/Venus}* mice, we observed that most of the CD4⁻CD8⁻ DN thymocytes expressed low levels of SATB1-Venus (Figs 1B and S1F). Further analyses of DN sub-populations revealed that SATB1-Venus is initially lowly expressed in CD44⁺CD25⁻ DN1 thymocytes, and it is then incrementally increased with DN1 transitioning into CD44⁺CD25⁺ DN2 and CD44⁻CD25⁺ DN3 thymocyte

stages (Fig S1F). In CD25⁻CD44⁻ DN4 thymocytes, there were SATB1-Venus^hi and SATB1-Venus^lo populations, the latter of which is likely to be non-T lymphoid cells. There was also a significant increase in the number of SATB1-Venus+ DP thymocytes, during their transition from CD4⁻CD8⁻ DN to CD4⁺CD8⁺ DP stage (Fig 1B). Notably, DP thymocytes undergoing positive selection (CD69⁺TCRβ^mid and CD69⁺TCRβ^hi) showed further increase in SATB1-Venus expression (Fig S1G), which may suggest that SATB1 expression, is positively controlled by TCR stimulation. In addition, there was a significant decline in SATB1-Venus expression from positively selected DPs to mature thymocytes (Figs 1B and S1G). Moreover, analyses of Satb1 transcript levels in DN3, DP, CD4 SP, and CD8 SP thymocytes from the immunological genome project (ImmGen, https://www.immgen.org/) (19) were highly consistent with the SATB1-Venus protein levels (Fig 1B). These data suggest that our *Satb1-Venus* reporter mouse model can be used to endogenously study the SATB1 protein function. In thymic Tregs, defined as CD24^loTCRβ^hiCD4⁺CD8α⁻CD25⁺ cells, SATB1-Venus expression was lower than their conventional CD4 counterparts (Fig 1C). Further analyses of nonconventional T cell subsets, such as the invariant natural killer T (iNKTs) and γδ-T cells, showed that iNKTs cells expressed similar levels of SATB1-Venus to that in conventional CD4 SP cells but was significantly higher than in γδ-T cells. (Fig S1H).

We next analysed SATB1-Venus expression in peripheral lymphocytes in the spleen. Both CD4⁺ and CD8⁺ T cells in the spleen showed higher SATB1-Venus expression than B cells (Fig 1D), which again showed high consistency with the Satb1 transcript levels (https://www.immgen.org/) (19). When CD4⁺ T cells were further gated for naïve (CD62L^hi CD44^lo), central memory (CD62L^hiCD44^hi), and effector memory (CD62L^loCD44^hi) populations, we noted a uniform and high SATB1-Venus expression in naïve CD4 T cells (Figs 1E and S1I). On the other hand, there were SATB1-Venus^hi and SATB1-Venus^lo populations in both CD4 central memory and CD4 effector memory populations, with increased frequency of SATB1-Venus^lo in effector memory cells. In the CD8 T cell pool, both naïve and central memory populations sustained a uniform and high SATB1-Venus expression, whereas the effector memory T cells consisted of both SATB1-Venus^hi and SATB1-Venus^lo populations. These observations suggest that T cell memory differentiation in the periphery induces down-regulation of SATB1 expression.

Intestinal intraepithelial lymphocytes (IEL) are most likely to represent effector T cells, as they are continuously exposed to various antigens in the gut. In contrast to splenic T cells of *Satb1^{+/Venus}* mice, 85% of which are SATB1-Venus^hi cells, only 2.5% of the intestinal TCRβ⁺ IEL were SATB1-Venus^hi (Fig 1F). This SATB1-Venus^hi IEL population expressed CD62L and showed similar CD4 and CD8α expression profiles to that in SATB1-Venus^hi splenic T cells. These data suggest that these CD62L⁺SATB1-Venus^hiTCRβ⁺ IEL are recent immigrants of circulating αβT cells. On the contrary, the

---

or CD8 T cell subpopulations: CD62L^hiCD44^lo naïve, CD62L^hiCD44^hi central memory, and CD62L^loCD44^hi effector memory T cells from *Satb1^{+/Venus}* mice. Numerical values indicate SATB1-Venus MFI and data are a representative of four biological experiments. Data are summarised in the adjacent graph. **(F)** Contour plots show SATB1-Venus and CD62L expression in splenic versus intraepithelial TCRβ⁺ T cells of *Satb1^{+/Venus}* mice, and the CD4 and CD8α expression patterns on CD62L⁻ SATB1-Venus^lo and CD62L⁺SATB1-Venus^hi T cell populations. Data are representative of three biological experiments. **(G)** Histograms show SATB1-Venus expression in naïve CD4 T cells undergoing activation in vitro for 0, 24, 48, and 72 h and summarised in the adjacent graph from three biological experiments. Statistics were calculated by unpaired *t* test; *$P$- < 0.05, **$P$- < 0.01, ***$P$- < 0.001. **(H)** Histograms show SATB1-Venus expression in naïve CD4 T cells versus in vitro differentiated CD4 $T_H0$, $T_H1$, $T_H2$, $T_H17$, and i$T_{Reg}$ cells from Satb1^{+/Venus} mice. Data are representative of three biological experiments and summarised in the adjacent graph. Statistics were calculated by two-way ANNOVA Tukey's multiple comparisons: *$P$- < 0.05, **$P$- < 0.01, ***$P$- < 0.001.

CD62L⁻SATB1-Venus^lo TCRβ⁺ IEL mainly represented a mixture of CD4⁺, CD4⁺CD8α⁺ and CD4⁻CD8α⁺ subpopulations and most of the CD4⁺CD8α⁺ cells were nonconventional CD8aα⁺ IELs (data not shown). This, therefore, suggests that SATB1-Venus could serve as a good marker for separating effector and naïve T cells.

To examine SATB1 expression during T cell activation, SATB1-Venus was quantified in $Satb1^{+/Venus}$ naïve CD4 T cells undergoing TCR activation in vitro. Indeed, SATB1-Venus levels significantly elevated after 24- and 48-h post activation and plateaued after 72 h post activation (Fig 1G). Furthermore, we traced SATB1-Venus expression in various effector CD4 helper T cell subsets that were differentiated in culture. Interestingly, SATB1-Venus expression is highest in CD4 $T_H1$, $T_H2$, and $T_H17$ and lowest in Tregs (Fig 1H). Overall, SATB1 expression in T cells is dynamically regulated during thymocyte development and in their functional differentiation into effector cells, in a cell-context-dependent manner.

## Identification of *cis*-regulatory genomic regions for *Satb1* gene regulation

Our *Satb1-Venus* reporter mice revealed that the amount of SATB1 protein is significantly altered during T-cell development and transcriptome data generated by the ImmGen project (https://www.immgen.org/) (19) shows a nice correlation of SATB1-Venus expression with *Satb1* mRNA levels. This indicates that amount of SATB1 protein is mainly regulated at the transcription level, rather than via post-translational mechanisms. Besides the usage of alternative gene promoters, cell-type specific gene expression is often regulated by *cis*-regulatory elements such as enhancer(s) (20), which are often located far from the transcription start site and shows higher chromatin accessibility (21). Therefore, we used publicly available ATAC-seq data (19) to search for open chromatin regions around the *Satb1* locus. There are two genomic regions upstream of the *Satb1* gene, which we referred them to as *Satb1-a* and *Satb1-b*, respectively, that showed significant ATAC-seq peaks in T cell subsets with activated/memory characteristics (Fig 2A, open chromatin regions highlighted in red dashed boxes). Sequences within these two genomic regions are evolutionally conserved, which imply that these regions may have some functions. To examine whether *Satb1-a* and *Satb1-b* genomic regions exhibit enhancer-like signatures, we also utilised publicly available H3K27 acetylation (H3K27Ac) ChIP-seq data (8), and indeed found that *Satb1-a* was much more highly enriched with H3K27Ac modifications than *Satb1-b*. This finding suggested that *Satb1-a* may function as an enhancer. Interestingly, the noncoding RNAs, *Gm19585*, and *Gm20098*, are transcribed near the *Satb1-a* and *Satb1-b* regions, respectively. RNA-seq data from ImmGen also showed that *Gm19585* and *Gm20098* are expressed in various T-cell subpopulations (Fig S2A). Specifically, *Gm19585* and *Satb1* transcripts were mostly co-expressed in various T cell populations, whereas *Gm20098* co-expression with *Satb1* is only limited to a small number of T cell subsets such as DP thymocytes. These observations prompted us to further study whether *Satb1-a* and *Satb1-b* function in *Satb1* gene regulation.

To determine whether the *Satb1-a* and *Satb1-b* regions have any roles in *Satb1* gene regulation, we used CRISPR/Cas9-mediated genome editing technology to delete the core sequences of

*Satb1-a* or *Satb1-b*, either on the $Satb1^{+}$ or $Satb1^{Venus}$ alleles. Two single-guide RNAs (sgRNAs) for *Satb1-a* or *Satb1-b* were selected and co-injected with the Cas9-encoding mRNA into $Satb1^{+/Venus}$ murine zygotes. The F0 founder mice that had deleted *Satb1-a* or *Satb1-b* regions were crossed with C57/B6N mice, and F1 founders carrying the $Satb1^{Venus}$ allele, together with the deletion of *Satb1-a* or *Satb1-b*, were selected as heterozygous mice with $Satb1^{+/Venus-Δa}$ or $Satb1^{+/Venus-Δb}$ genotype. Among two and three F1 $Satb1^{+/Venus-Δa}$ and $Satb1^{+/Venus-Δb}$ founders, we chose one line as a representative for $Satb1^{Venus-Δa}$ and $Satb1^{Venus-Δb}$ (Sequence information shown in Fig S2B) and were examined for the expression of SATB1-Venus in various T cell subsets. In both $Satb1^{+/Venus-Δa}$ and $Satb1^{+/Venus-Δb}$ mice, there were no significant changes in SATB1-Venus expression in all DN, DP, CD4-SP, and CD8-SP thymocytes (Fig S2C). Analyses of spleens, however, revealed that SATB1-Venus expression levels were significantly reduced in naïve CD4 T cells of $Satb1^{+/Venus-Δa}$, which were not observed in $Satb1^{+/Venus-Δb}$ mutant mice (Fig 2B). It was also noteworthy that loss of *Satb1-a* caused a significant reduction in SATB1-Venus expression in CD4 effector-memory populations but had no effects on CD8 naïve and memory T cell populations. These results suggest that *Satb1-a* is essential for maintaining *Satb1* expression in peripheral CD4 T cells with activated/memory phenotypes.

To elucidate whether *Satb1-a* and *Satb1-b* have any roles in regulating SATB1 expression in effector CD4 T cell subsets, we activated naïve CD4 T cells of $Satb1^{+/Venus-Δa}$ and $Satb1^{+/Venus-Δb}$ mutant mice in vitro, under $T_H0$, $T_H1$, $T_H2$, $T_H17$ or $iT_{reg}$ polarising conditions. Interestingly, we observed a substantial reduction in SATB1-Venus expression in $Satb1^{+/Venus-Δa}$ CD4 $T_H0$ and $T_H17$ cells, but most strikingly, SATB1-Venus expression was almost lost in $Satb1^{+/Venus-Δa}$ $T_H2$ cells (Fig 2C). Therefore, the *Satb1-a* enhancer may allow to maintain high levels of SATB1 expression in $T_H2$ cells. On the contrary, we did not detect any decline in SATB1-Venus in $Satb1^{+/Venus-Δb}$ CD4 $T_H0$, $T_H1$, $T_H2$, $T_H17$ or $iT_{reg}$ cells (Fig 2C), thereby discarding any potential roles of *Satb1-b* in regulating *Satb1* expression in T cells. We also confirmed that removal of the *Satb1-a* region from the *Satb1* locus resulted in a significant reduction of SATB1 protein levels in in vitro-differentiated CD4 $T_H2$ and $T_H0$ cells (Fig S2D), which was accompanied with reduction of *Satb1* transcripts (Fig 2D). These results confirm that *Satb1-a* is essential to maintain *Satb1* expression under $T_H2$ polarising conditions.

### *Satb1-a* region functions as a genomic enhancer

In addition to the decline of *Satb1* expression, we also observed a significant decline in the expression of *Gm19585* transcripts in $Satb1^{Δa/Δa}$ CD4 $T_H2$ cells (Fig 2D). Therefore, one would speculate whether the loss of *Gm19585* transcripts, which may function as noncoding RNAs, were causative for impaired *Satb1* expression. Hence, to mechanistically understand how *Satb1-a* controls *cis* expression of *Satb1* in CD4 $T_H2$ cells, we examined whether the noncoding *Gm19585* RNAs are primarily involved in stabilising *Satb1* expression. To this aim, we performed *Gm19585* knockdown studies in $T_H2$-polarised CD4 T cells in vitro. Analyses of *Gm19585* transcripts deposited on UCSC genome browser showed that there are at least eight alternatively spliced transcripts, as some transcripts were either exon1- or exon2-depleted sequences. Therefore, three shRNA

**Figure 2. Analyses of *Satb1-a* and *Satb1-b* function in regulating *Satb1* expression.**
**(A)** ATAC-seq and H3K27Ac ChIP-seq signals in *Satb1*, and in upstream genomic regions transcribing *Gm20098* and *Gm19585* in various murine T cell populations indicated. Red dashed boxes highlight genomic regions, *Satb1-a* and *Satb1-b*, that show ATAC-seq and/or H3K27Ac ChIP-seq signals in thymocytes and/or T cells. *Satb1-a* and *Satb1-b* are evolutionarily conserved and located near regions transcribing *Gm19585* and *Gm20098*, respectively. **(B)** Histograms show SATB1-Venus expression in

expression plasmids, *pLKO-Gfp-Gm19585-1*, *pLKO-Gfp-Gm19585-2*, and *pLKO-Gfp-Gm19585-4*, that target exon1, exon2, and exon4 of *Gm19585* transcript respectively, were generated (Fig S2E). In vitro-activated CD4 T cells were transduced with these shRNAs expressing lentiviral particles and were maintained in $T_H2$ polarising conditions for additional 6 d. shRNA-transduced CD4 $T_H2$ cells defined as CD4⁺GFP⁺ were sorted and were subjected to *Satb1* transcriptional analyses by qRT–PCR. *pLKO-Gfp-Gm19585-2* or *pLKO-Gfp-Gm19585-4* shRNAs were able to knockdown *Gm19585* transcripts by twofold but had no significant effects on *Satb1* transcript levels in CD4 $T_H2$ cells (Fig 3A). These results suggest that the noncoding RNAs from *Gm19585* are unlikely to be responsible for maintaining *Satb1* expression under $T_H2$ polarising conditions.

The above finding raised the possibility that *Satb1-a* functions as an enhancer to control *Satb1* gene expression in *cis*. To investigate this possibility, we used a publicly available three-enzyme Hi-C (3e Hi-C) dataset from murine embryonic stem cells, thymocytes, and CD4 $T_H2$ cells (22, 23), to identify genome wide interactions between *Satb1*-exon1 (which contain promoters) and *Satb1-a* region. Interestingly, we found that *Satb1*-exon1 and *Satb1-a* are closely located within a topologically associated domain, in both thymocytes and CD4 $T_H2$ cells (Fig 3B). This indicates that a specific *cis*-genomic interaction between these regions is induced specifically in T cells, through chromatin looping. For further validation, we performed a chromatin capture conformation (3C) assay to confirm whether this chromatin interaction between *Satb1*-exon1 and *Satb1-a* specifically occurs in CD4 $T_H2$ cells. For this study, we designed an anchor primer AS2 for capturing *Satb1-a* and one primer S1 to capture *Satb1 exon1b*, an alternative *Satb1* promoter that has been reported to be highly expressed in CD4 $T_H2$ cells (15) (Fig 3C); the primers AS1 and S2 were used as negative controls. As expected, the use of AS2-S1 primers strongly produced significant amplicon signals in CD4 $T_H2$ cells than in CD4 $T_H0$ and $T_H1$ cells. Therefore, our 3C assay validated that the chromatin interaction between *Satb1*-exon1 and *Satb1-a* genomic regions is formed in CD4 $T_H2$ cells, to regulate *Satb1* gene expression.

Given that the regulation of chromatin looping is often mediated by a transcription factor, we next sought to identify which $T_H2$-specific transcription factor could be responsible for promoting chromatin looping of *Satb1-a* to the *Satb1* promoters in CD4 $T_H2$ cells. It is already established that the signal transducer and activator of transcription 6 (STAT6) is activated downstream of IL-4 signalling in CD4 $T_H2$ cells and directly regulates *Satb1* expression (15). We noticed that there are STAT DNA-binding motifs within the *Satb1-a* region. Hence, we examined publicly available STAT6 ChIP-seq datasets in murine CD4 $T_H2$ cells (24) and found that *Satb1-a* was indeed occupied by STAT6 in these cells, suggesting that STAT6 binds and activates the *Satb1-a* in $T_H2$ cells (Fig 3D). Therefore, we

have classified *Satb1-a* as an enhancer of *Satb1* expression in the context of $T_H2$ and henceforth shall be referred as *Satb1-Eth2* (enhancer for $T_H2$ cells).

### *Satb1-Eth2* is essential in repressing IL-5 expression in CD4 $T_H2$ cells in vitro

Having established that *Satb1-Eth2* is essential in maintaining *Satb1* expression in CD4 $T_H2$ cells, we then investigated the relevance of *Satb1-Eth2* on the differentiation and function of CD4 $T_H2$ cells. For this aim, we established *Satb1^{ΔEth2/ΔEth2}* mice on C57BL6 background by genome editing. First, it was previously reported that SATB1 functions to promote *Gata3* expression in CD4 $T_H2$ cells in vitro (14). Therefore, we first considered whether *Satb1-Eth2* had any role in regulating *Gata3* expression in CD4 $T_H2$ cells in vitro. Interestingly, although *Satb1^{ΔEth2/ΔEth2}* CD4⁺ $T_H2$ cells showed a significant loss of *Satb1* mRNA, there was no significant reduction in *Gata3* expression (Fig S3A). These data suggest that low levels of *Satb1*, via loss of *Satb1-Eth2*, did not impair *Gata3* expression and primary CD4 $T_H2$ differentiation in vitro. Next, it has been reported that SATB1 binds and regulates the expression of the $T_H2$ cytokines IL-4, IL-5, and IL-13 (12), and hence, we examined whether *Satb1-Eth2* is essential for the expression of IL-4, IL-5, and IL-13 in *Satb1^{ΔEth2/ΔEth2}* CD4 $T_H2$ cells in vitro. PMA and ionomycin stimulation in *Satb1^{ΔEth2/ΔEth2}* CD4 $T_H2$ cells revealed no significant differences in the intracellular expression of IL-4, which suggested that *Satb1-Eth2* is not required for IL-4 induction in CD4 $T_H2$ cells (Fig S3B). We, however, did find a significant increase in IL4⁺IL-5⁺ cells in our *Satb1^{ΔEth2/ΔEth2}* CD4 $T_H2$ cultures (Fig 4A). The increase in IL4⁺IL-5⁺ cells in our *Satb1^{ΔEth2/ΔEth2}* CD4 $T_H2$ cultures was also recaptured upon TCR stimulation (Fig S3C). Further analyses of *Il-4*, *Il-5*, and *Il-13* transcripts in PMA and ionomycin-stimulated *Satb1^{ΔEth2/ΔEth2}* CD4 $T_H2$ cells revealed that only *Il-5* transcripts were significantly elevated in *Satb1^{ΔEth2/ΔEth2}* CD4 $T_H2$ cells (Fig S3D). Moreover, the elevated expression of *Il-5* in *Satb1^{ΔEth2/ΔEth2}* CD4 $T_H2$ cells coincided with elevated IL-5 secretion in culture supernatants post PMA and ionomycin stimulation (Fig 4B). These results collectively show that *Satb1-Eth2* is crucial to maintain SATB1 levels in CD4 $T_H2$ cells, and to restrain IL-5 expression levels.

### *Satb1-Eth2* is required to maintain SATB1 expression in CD4 $T_H2$ and ILC2s during $T_H2$ immune responses in vivo

We have demonstrated that *Satb1^{ΔEth2/ΔEth2}* CD4 $T_H2$ cells have elevated the expression of IL-5 in vitro. These results are consistent with former studies showing the requirement of SATB1 in suppressing IL-5 in human CD4 $T_H2$ cells (13). IL-5 is the key soluble factor for eosinophil activation and recruitment. At the site of a $T_H2$

splenic CD4 or CD8 T cell subpopulations (1. CD62L^{hi}CD44^{lo} naïve, 2. CD62L^{hi}CD44^{hi} central memory, and 3.CD62L^{lo}CD44^{hi} effector memory T cells) from *Satb1^{+/Venus}*, *Satb1^{+/Venus-Δa}*, and *Satb1^{+/Venus-Δb}* mutant mice. Numerical values indicate SATB1-Venus MFI (mean fluorescence intensity). Graph shows quantification of SATB1-Venus MFI in splenic CD4 and CD8 T cell subpopulations. Data are summarised from four biological experiments. Statistics were calculated by two-way ANNOVA Tukey`s multiple comparisons; *P- < 0.05, **P- < 0.01, ***P- < 0.001. **(C)** Histograms show SATB1-Venus expression in in vitro differentiated CD4 $T_H0$, $T_H1$, $T_H2$, $T_H17$, and iT_{Reg} cells, generated from *Satb1^{Venus-Δa}* (top, n = 3) and *Satb1^{Venus-Δb}* (bottom, n = 2) mutant mice. Data are a representative of at least two biological experiments. Adjacent graphs show quantification of SATB1-Venus MFI. Data are summarised from two or three biological experiments. **(D)** qRT–PCR analysis of *Satb1* and *Gm19585* expression in *Satb1^{+/+}* and *Satb1^{Δa/Δa}* CD4 $T_H0$, $T_H1$, $T_H2$ cells. Data are summarised from four biological experiments. Statistics were calculated by unpaired *t* test; *P- < 0.05, **P- < 0.01, ***P- < 0.001, ****P- < 0.0001.

**Figure 3. *Satb1-a* interacts with *Satb1* promoter and is bound by STAT6.**
**(A)** Graphs show qRT–PCR analyses of *Gm19585* and *Satb1* expression in CD4 T$_H$2 cells transduced with *pLKO-GFP-empty*, *pLKO-Gfp-Gm19585-*1, *pLKO-Gfp-Gm19585-*2 or *pLKO-Gfp-Gm19585-*4 shRNA-expressing vectors. Data are representative of at least two biological experiments. **(B)** Heatmaps show contact matrices of chromosome 17 from murine embryonic stem cells (top) versus thymocytes (middle) and CD4 T$_H$2 cells (bottom). The location of *Satb1-exon1* and *Satb1-a* are pinpointed by black arrows. **(C)** Schematic shows the locations of *Satb1-exon1* and *Satb1-a* loci and primer design for the 3C assay. The positions of primers S1 and S2 overlap Satb1-exon1b and Satb1-exon1a, respectively. The position of anchor primers AS1 and AS2, respectively, align outside and nearby the *Satb1-a* region. The primer pairs S1-AS1, S1-AS2, S2-AS1, and S2-AS2 were used in combination to capture interactions between *Satb1-exon1* and *Satb1-a*. Data are summarised in the graph below, from two biological experiments. **(D)** Genome browser ChIP-seq tracks show binding of STAT6 to the *Satb1-a* in murine CD4 T$_H$2 cells. ATAC-seq signals in colonic Tregs are shown as reference. Two putative STAT6-binding motifs within the *Satb1-a* region are indicated as grey ovals.

immune response, eosinophils release granules containing proinflammatory proteins that cause inflammation and tissue damage. Dysregulation in IL-5 expression and eosinophil activity drive allergic reactions and have clinical implications in the progression of asthmatic responses. We therefore speculated that SATB1 functions to repress CD4 T$_H$2 immune responses via IL-5 suppression and promotes immune resolution. Thus, we next explored the in vivo function of *Satb1-Eth2* in controlling type 2 immune responses and used an extract of *Alternaria alternata* (*A.A.*) to induce experimental airway inflammation. First, we traced eosinophils infiltration in the bronchial alveolar lavage (BAL) after treating the *Satb1*$^{+/Venus}$ mice with *A.A.* on days 4, 7, and 10, and confirmed it to peak from day 7 to 10 (Fig S4A). On day 7, we found a small but significant induction of SATB1-Venus in CD4 T$_H$2 cells (Fig S4B), which is further maintained at day 10. During this study, we also noticed that SATB1-Venus expression was induced in ILC2s (Fig

S4B), a subset of lymphocytes responsible for T$_H$2 responses at the early phase of lung inflammation. Previous studies failed to detect *Satb1* expression in ILC2s of naïve mice, so we suspected whether ILC2s could only induce *Satb1* expression during lung inflammation. Indeed, ex vivo analyses of lung resident ILC2s in naïve mice revealed very little expression of SATB1-Venus expression (Fig S4B). However, in vitro activation of lung-resident ILC2s with IL-33, IL-2, and IL-7 resulted in a significant induction of SATB1-Venus from *Satb1*$^{Venus/Venus}$ mice (Fig S4C). Importantly, this SATB1-Venus induction was completely absent in in vitro-activated ILC2s from *Satb1*$^{Venus-ΔEth2/Venus-ΔEth2}$ mice (Fig S4C). These results indicate that *Satb1-Eth2* is indispensable for *Satb1* induction in activated ILC2s. We next examined SATB1-Venus expression in lung CD4 T$_H$2 cells and ILC2s with or without the *Satb1-Eth2* enhancer 7–10 d post *A.A.* extract injection. To our expectations, BAL-CD4 T cells and lung-derived CD4 T$_H$2 cells and ILC2s from in *A.A.*-injected *Satb1*$^{+/Venus-ΔEth2}$

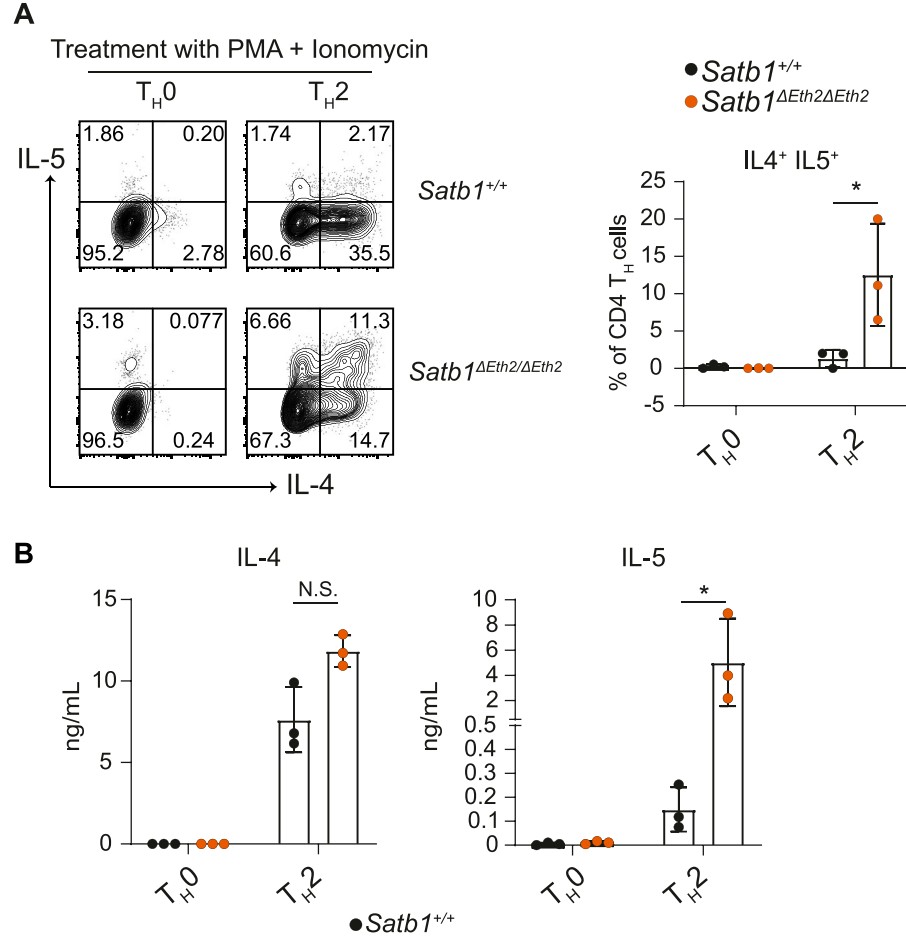

**A**

Treatment with PMA + Ionomycin

Figure 4.   Analyses of *Satb1-Eth2* (*Satb1-a*) in CD4 $T_H2$ differentiation and function in vitro.
**(A, B)** Intracellular expression of IL-4 and IL-5 (A) and their concentrations in culture supernatants (B) from PMA- and -onomycin-stimulated *Satb1⁺/⁺* and *Satb1^ΔEth2/ΔEth2^* in vitro-differentiated $T_H0$ and $T_H2$ cells. Data are representative of at least three biological experiments and are summarised in the adjacent graph. Statistics were calculated by unpaired *t* test; *$P$- < 0.05, **$P$- < 0.01, ***$P$- < 0.001.

**B**

mice failed to up-regulate SATB1-Venus expression (Fig 5A). Crucially, these results demonstrate the strict requirement of *Satb1-Eth2* in inducing SATB1 expression during in vivo $T_H2$ immune responses. To determine whether loss of *Satb1-Eth2* function exacerbated lung inflammation in response to *A.A.*, we examined eosinophil infiltration in BAL of *A.A.*-treated *Satb1^ΔEth2/ΔEth2^* mice versus *Satb1⁺/⁺* but found no significant increase in eosinophil numbers in *A.A.*-treated *Satb1 ^ΔEth2/ΔEth2^* mice at 7 d post-injection (Fig 5B). In addition, quantification of IL-5 in the BAL supernatants of *A.A.*-treated *Satb1^ΔEth2/ΔEth2^* mice also revealed no significant increase in IL-5 expression (Fig 5B). Alternatively, we also used the OVA induced lung inflammation model to induce $T_H2$ immune responses in *Satb1 ^ΔEth2/ΔEth2^* mice (Fig S4D) but again found no significant increase in eosinophil numbers and in IL-5 concentrations in the BAL of OVA-challenged *Satb1^ΔEth2/ΔEth2^* mice (Fig S4E). Lastly, sorted lung CD4 $T_H2$ cells and ILC2s from *A.A.*-treated mice, at 10 d post-injection, showed similar frequencies of IL-4⁺IL-5⁺ or IL4⁻IL-5⁺ subpopulations between *Satb1⁺/⁺* and *Satb1 ^ΔEth2/ΔEth2^* mice (Fig 5C). Hence, our in vivo models could not find a functional relevance of *Satb1-Eth2* in restraining both $T_H2$ immune responses and IL-5 expression in CD4 $T_H2$ cells and ILC2s during *A.A.*-induced lung inflammation. Overall, our results from the *A.A.*-induced

lung inflammation model confirmed that *Satb1-Eth2* is required to maintain SATB1 expression in innate and adaptive $T_H2$ lymphocytes during acute lung inflammation, but the physiological relevance of *Satb1-Eth2* in regulating $T_H2$ immune responses waits for further investigation.

# Discussion

A recent GWAS study identified a Psoriasis-associated single nucleotide polymorphism in *SATB1-AS1*, an antisense RNA that is encoded 240 kb upstream of the *SATB1* locus and showed a T-cell–specific interaction with the *SATB1* promoter (17). This study was not only the first to reveal the possibility of *cis*-regulatory transcriptional mechanisms that could regulate *Satb1* expression in both human and murine T cells, but it was also the first-link SATB1 to allergy and inflammatory disease.

Previous studies have examined the function of SATB1 in $T_H2$ cells. SATB1 was shown to bind to the murine $T_H2$ locus and promoted *Il-5*, *Il-4*, and *Il-13* gene expression (12). Another study showed the induction of *Satb1* expression upon IL-4 signalling in

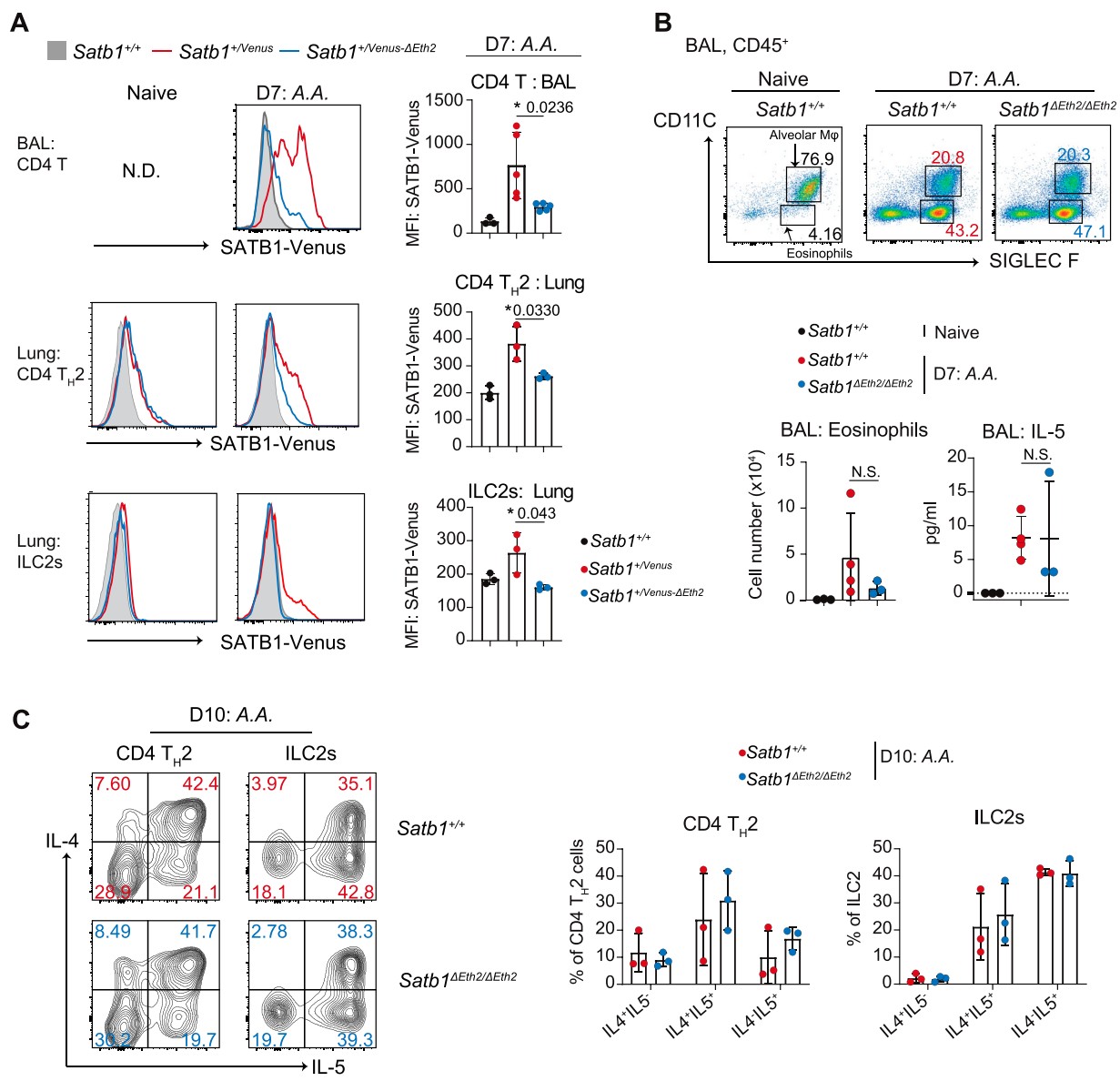

**Figure 5.  *Satb1-Eth2* functions to regulate SATB1 expression in CD4 T$_H$2 cells and activated ILC2s in vivo.**
**(A)** Histograms show SATB1-Venus expression in BAL CD4$^+$ T cells (CD45$^+$TCR$\beta^+$CD4$^+$), in lung CD4 T$_H$2 cells (CD45$^+$TCR$\beta^+$CD4$^+$GATA3$^+$ST2$^+$), and in lung ILC2s (CD45$^+$TCR$\beta^-$CD4$^-$GATA3$^+$ST2$^+$) of naïve and *A.A.* treated *Satb1$^{+/+}$*, *Satb1$^{+/Venus}$* and *Satb1$^{+/Venus-\Delta Eth2}$* mice on day 7. Data from *A.A.*-injected mice are a representative of three biological replicates. Graphs summarise SATB1-Venus expression in those cells from *A.A.* treated *Satb1$^{+/+}$*, *Satb1$^{+/Venus}$* and *Satb1$^{+/Venus-\Delta Eth2}$* mice on day 7. **(B)** Dot plots show frequencies of eosinophils in BAL of naïve (*Satb1$^{+/+}$*) and *A.A.*-treated *Satb1$^{+/+}$* and *Satb1$^{\Delta Eth2/\Delta Eth2}$* mice on day 7. Graphs summarise eosinophil numbers and IL-5 concentration in BAL of naïve (*Satb1$^{+/+}$*) and *A.A.*-treated *Satb1$^{+/+}$* and *Satb1$^{\Delta Eth2/\Delta Eth2}$* mice on day 7. Data are summarised from at least three biological replicates. **(C)** Contour plots show the intracellular expression of IL-4 and IL-5 in PMA and ionomycin-stimulated lung CD4 T$_H$2 cells and ILC2s, from *A.A.*-treated *Satb1$^{+/+}$* and *Satb1$^{\Delta Eth2/\Delta Eth2}$* mice on day 10. Graphs show the frequencies of IL-4$^+$IL-5$^-$, IL-4$^+$IL-5$^+$, and IL-4$^-$IL-5$^-$ subpopulations in PMA- and ionomycin-stimulated lung CD4 T$_H$2 cells and ILC2s from *A.A.*-treated *Satb1$^{+/+}$* and *Satb1$^{\Delta Eth2/\Delta Eth2}$* mice on day 10. Data are summarised from three biological replicates. Statistics were calculated by unpaired *t* test; *$P$- < 0.05, **$P$- < 0.01, ***$P$- < 0.001.

in vitro-differentiated CD4 T$_H$2 cells (15), in a STAT6-dependent manner. This up-regulation of *Satb1* expression in CD4 T$_H$2 cells is associated with altered *Satb1* promoter usage from P1 to P2 and P3, but the biological relevance of these promoters in regulating *Satb1* expression during T cell development were not fully addressed. Hence, it was unclear how *Satb1* expression is specifically controlled during effector CD4 T$_H$ cell differentiation, particularly in CD4 T$_H$2 cells. In addition, current studies have not examined whether *Satb1* was induced during activation of ILC2s, in part because of low *Satb1* expression in steady state ILC2s. Here, we have identified and shown a crucial function of *Satb1-Eth2* in regulating *Satb1* expression in T$_H$2 cells and ILC2s in vitro and in vivo. Loss of *Satb1-Eth2* function significantly impacted on the transcriptional levels of *Satb1*, specifically in T$_H$2 cells and activated

ILC2s. Interestingly, *Satb1-Eth2* overlapped with exon 4 of the noncoding gene, *Gm19585*, and loss of *Satb1-Eth2* function was accompanied with the reduction of *Gm19585* transcript levels. However, knockdown of *Gm19585* transcripts did not affect *Satb1* expression in T$_H$2 cells, which suggests that the *Gm19585* transcripts have no role in regulating *Satb1* gene expression. Rather, chromatin looping between *Satb1-Eth2* with *Satb1* promoters and STAT6 binding to the *Satb1-Eth2* in T$_H$2 cells support the notion that *Satb1-Eth2* functioned as an IL-4 inducible T$_H$2-specific enhancer for *Satb1* gene expression.

Loss of *Satb1-Eth2* expression significantly caused depression of IL-5 in T$_H$2 cells in vitro, which therefore highlights a biological role of *Satb1-Eth2* in suppressing IL-5 expression in T$_H$2 cells. Our study is also consistent with what others have observed with *Satb1* knockdown assays in human CD4 T$_H$2 cells ([13]), and therefore indicates that IL-5 de-repression, caused by loss of the *Satb1-Eth2* function, is mediated by the reduction of SATB1. Notably, IL-5 is known to play a major role in promoting eosinophil recruitment and promote the progression of allergy or asthma ([25]). A previous report presumed that SATB1 had a significant role in suppressing lung inflammation, but this was never examined in in vivo models ([13]). Indeed, we examined the role of *Satb1-Eth2* in augmenting lung inflammation in vivo but found that loss of the *Satb1-Eth2* had no additive effects on eosinophil infiltration or on IL-5 expression in the BAL of *A.A.*-treated mice, albeit of low *Satb1* expression in both lung T$_H$2 cells and ILC2s. Based on this key information, one would have to truly consider whether SATB1 has any function in controlling T$_H$2 immune responses in vivo. However, these results do not formally exclude the involvement of SATB1 in controlling T$_H$2 immune responses in vivo under different experimental settings, for instance in chronic lung inflammation and in atopic-dermatitis models ([26]), and thus would merit further investigation.

We also uncovered a partial but significant reduction of SATB1-Venus expression in naïve and effector memory CD4 T cells, caused by loss of *Satb1-Eth2* function. Naïve CD4 T cells undergo homeostatic proliferation to maintain their survival in the periphery, which crucially requires both self-MHC ligands and the IL-7 cytokine ([27]). In addition, effector memory T cells require IL-15 for their survival. Both IL-7 and IL-15 signalling pathways activate the STAT5 transcription factors downstream. It should be noted that most of the STAT transcription factor members recognise the palindromic DNA sequence TTCNNNGAA. It is therefore possible that, in both naïve and effector memory CD4 T cells, STAT5 could bind to *Satb1-Eth2* and regulate *Satb1* expression. However, because loss of *Satb1-Eth2* function had a partial effect on SATB1 expression in naïve CD4 T cells, we believe that *Satb1-Eth2* is partially responsive to either IL-7 and/or IL-15. Alternatively, the regulation of *Satb1* expression in naïve and memory CD4 T cells could be mediated by other *cis-* or *trans-* enhancers, which may require an in-depth analysis.

Overall, the present study identifies and demonstrates the essential role of the *cis*-regulatory enhancer, *Satb1-Eth2,* in regulating *Satb1* expression, specifically in CD4 T$_H$2 cells and ILC2s, which significantly contributes to our current understandings of how *Satb1* expression is controlled during T$_H$2-mediated immune responses. Our study signifies the beginnings of an *era* of identifying *cis-* and/or *trans*-regulatory elements that regulate *Satb1* gene

expression in T cells and thus provides insightful information on *Satb1* function in a T cell context-dependent manner.

# Materials and Methods

### Mice

The *Satb1$^{Venus}$* allele was generated by knock-in insertion of the Venus open reading frame downstream of transcription start site (exon2) of the *Satb1* gene. For this aim, a BAC clone B6Ng01-312L13, which included the 5′ area of the *Satb1* gene, was purchased from RIKEN BRC. A 13-kb genomic region harboring the ATG start codon was subcloned into the pBlueScript II vector (Stratagene). The DNA fragment harboring partial exon2 sequences and Venus cDNA sequences were generated by overlapping the PCR technique and was used to construct a targeting vector. The targeting vector was transfected into murine M1 ES cells as previously described ([28]). G418-resistant ES clones were screened for homologous recombination event between the target vector and the *Satb1* gene. Appropriate ES clones were then used to generate chimera mice by ES aggregation. F1 founders from the chimera mice carrying the *Satb1$^{Venus}$* allele were selected for establishing mouse line and were analysed. *Satb1$^{Flx}$* CD4-*cre* mice were previously described in ([7]). *Satb1$^{+/Venus-Δa}$* (or *Satb1$^{Venus-ΔEth2}$*) and *Satb1$^{+/Venus-Δb}$* mice were generated by co-injection of Cas9-mRNA with sgRNAs into *Satb1$^{+/Venus}$* fertilised eggs that were generated by in vitro fertilisation with sperm from a *Satb1*$^{Venus/Venus}$ male mouse and *Satb1$^{+/+}$* oocytes. The F0 founders that had deleted *Satb1-a* or *Satb1-b* regions were then crossed with C57/B6N mice and F1 founders that deleted *Satb1-a* or *Satb1-b* on the *Satb1$^{Venus}$* allele were selected as heterozygous mice with the *Satb1$^{+/Venus-Δa}$* or *Satb1$^{+/Venus-Δb}$* genotype. Among the two and three *F1 Satb1$^{+/Venus-Δa}$* and *Satb1$^{+/Venus-Δb}$* founders, we chose one line as a representative for *Satb1$^{Venus-Δa}$* and *Satb1 $^{Venus-Δb}$* and were examined with littermate controls. Similarly, C57/B6N *Satb1$^{Δa}$* (*Satb1$^{ΔEth2}$*) mice were generated by co-injection of Cas9-mRNA with *Satb1-a* gRNA into C57Bl/6 fertilised eggs. Sequences for sgRNA are as follows: *Satb1$^{Δa}$* sgRNAs 5′-TCAACATCAGAATTTCT-3′ and 5′-CAGTCAACATCAGAATTTCT-3′; and *Satb1$^{Δb}$* sgRNAs 5′-ACACACACTGTCTGTTGTGC-3′ and 5′-GCTGCCTGCTTTTACATATC-3′. All mice were maintained at the RIKEN Center for Integrative Medical Sciences. The animal protocol was approved by the Institutional Animal Care and Use Committee of RIKEN Yokohama Branch (2020-026). For all experimental procedures, at least two to three mice were used per experiment and are shown either as a representative or as a summary of these biological experiments.

### Immunohistochemistry

50,000 of thymocytes from *Satb1$^{+/Venus}$* mice were resuspended in 0.1 ml RPMI-1640 medium (10% FBS) and were mounted on a poly-L-lysine-coated glass slide (REF: 26414; Polysciences, Inc.) and incubated at 37°C for 2 h. The attached cells were gently washed three times in PBS. The cells were fixed with 4% PFA at room temperature for 10 min and then washed three times in PBS. The cells were

permeabilized with 70% ethanol and kept at –20°C until analysis. On the day of the analysis, thymocytes were stained with 1/100 of DAPI (ab228549; Abcam) at room temperature, for 30 min, and washed three times in PBS. Cover slips were mounted on top of the thymocytes with Fluoromount-G mounting medium (00-4958-02; Thermo Fisher Scientific) and sealed with Biotium CoverGrip Coverslip Sealant (23005; REF). Confocal microscopic images were obtained with TCS-SP5 (Leica Microsystems) using 40x objective.

**Ex vivo cell preparation for flow cytometry**

Thymus, spleen, and peripheral lymph nodes (axillary, inguinal, and cervical) were removed from mice at 4–8 wk of age and were mashed through a 70-$\mu$m cell strainer in a Petri dish to make single-cell suspensions. For the elimination of red blood cells, splenocytes were treated with ACK lysis buffer (Gibco A1049201, 2.5 ml per spleen) for 3 min and pelleted by centrifugation at 300$g$ at 4°C for 5 min. Supernatants were discarded and cell pellets were resuspended into RPMI-1640 (2% FBS). All cell suspensions were maintained at 4°C for flow cytometry analyses.

For the preparation of IEL, the small intestine was cut into three sections and Peyer Patches were removed. Each section was cut longitudinally and washed twice with HBSS (5% of FCS). Sections were further cut into 0.5-cm pieces and put into a 50 ml conical tube containing 25 ml HBSS (5% of FCS). The tissues were then placed in a bioshaker (TAITEC, BR-43FL) at 37°C, at 250 rpm for 15 min. The supernatant was transferred into a new 50 ml tube, through a metal mesh, and cells were pelleted down by centrifugation at 300$g$ at 4°C for 5 min. The supernatant was discarded. The cell pellet was then resuspended in 8.5 ml of HBSS + 40% Percoll and layered onto 2 ml HBSS+ 70% Percoll in a 15 ml falcon tube. The cells were then centrifuged at 860$g$ for 25 min at room temperature with brakes off. IEL were collected in the interphase between 40% and 70% Percoll and washed twice with HBSS (5% of FCS). IEL were then subjected to flow cytometric analyses.

For the preparation of lung cell suspensions, the lungs were first perfused with 20 ml of PBS through the left ventricle of the heart. Perfused lungs were dissected out and minced up before incubation in RPMI-1640 (2% FBS) + 0.3 mg/ml collagenase IV (C-5138; Sigma-Aldrich) + 0.3 mg/ml of DNase I (043-26773; Wako) at 37°C for 45 min with continuous shaking. To homogenize the samples, the digested lung tissues were mashed in a 70-$\mu$m cell strainer in a Petri dish and cell suspensions were then pelleted by centrifugation at 300$g$ at 4°C for 5 min. For the elimination of debris, the cell pellets were resuspended in 10 ml RPMI (2% FBS) + 30% Percoll and separated by centrifugation at 860$g$ at room temperature for 20 min with brakes off. Debris in the upper phase were aspirated off and cell pellets were resuspended in 5 ml RPMI-1640 (2% FBS). The cells were washed and pelleted by centrifugation at 300$g$ at 4°C for 5 min and were then subjected for analysis.

**Flow cytometry**

Cells were stained with the following fluorophore-conjugated antibodies: B220 (RA3-6B2) CD4 (RM4-5), CD8a (53-6.7), CD11B (M1/70), CD11C (HL3), CD19 (1D3), CD24 (M1/69), CD25 (PC61.5), CD44 (IM7), CD45 (30-F11), CD62L (MEL-14), CD69 (H1.2F3), FOXP3 (FJK-16s), GATA3 (L50-

823), Gr1 (RB6-8C5), IL-4 (11B11), IL-5 (TRFK5), IFN-$\gamma$ (XMG1.2), Ly6C (AL-21), Ly6G (1A8), NK1.1 (PK136) Siglec-F (E13-161.7), ST2 (U29-93) TCR$\beta$ (H57-597), TCR$\gamma\delta$ (GL3), and Ter119 (TER-119). Murine CD1d dimmer XI (557599; BD-Bioscience) loaded with $\alpha$-GalCer was used to define iNKT cells. For flow cytometric analyses 1–2 × 10$^6$ cells were washed out of complete medium, by centrifugation at 300$g$ at 4°C for 5 min and resuspended in FACS buffer (PBS + 1% FBS and 0.05% NaN$_3$). For the analyses of extracellular marker on live cells, cell suspensions from spleen, pLNs, IEL, and lungs were treated with rat $\alpha$-mouse CD16/CD32 (BD Biosciences) for 10 min at 4°C. Fluorophore antibodies for extracellular markers were then added and stained for 30 min at 4°C. After incubation, cells were then pelleted by centrifugation at 300$g$ at 4°C for 5 min and resuspended in FACS buffer containing 7AAD/DAPI cell viability dye. For analyses of both extracellular and intracellular markers, cell suspensions of all tissues were stained with fixable live/dead cells (65-0866-18; Ebiosciences) for 10 min before rat $\alpha$-mouse CD16/CD32 treatment and extracellular marker staining. Using the FOXP3 staining buffer kit (00-5523-00; Ebiosciences), stained cells were then fixed, permeabilised, and stained for intracellular transcription factors as per the manufacturer's instructions. All cytometry analyses were performed using a FACS CANTO II (BD-Bioscience) and data were analysed using FlowJo (Tree Star) software.

**In vitro CD4 T helper differentiation**

Naïve CD4 T cells were pooled from spleen and peripheral lymph nodes. The spleen and peripheral lymph nodes (axillary, inguinal, and cervical) were removed from mice at 6–10 wk of age and were mashed through a 70-$\mu$m cell strainer in a Petri dish to make single-cell suspensions. Splenocytes were the treated with ACK lysis buffer (Gibco A1049201, 2.5 ml per spleen) for 3 min before pelleting (by centrifugation at 300$g$ at 4°C for 5 min) and resuspension into RPMI-1640 (2% FBS). Naïve CD4 T cells from either sorted by FACS Aria (sorted for CD4$^+$CD25$^-$CD62L$^{hi}$CD44$^+$) or isolated by using EasySep mouse naïve CD4 T cell isolation kit (ST-19765). Naïve T cells were resuspended in complete medium (KOHJIN BIO DMEM-H [16003016] + 10% FBS) and activated in the presence of 2 $\mu$g/ml of anti-CD3 (REF: 553058; BD Pharmingen) and 2 $\mu$g/ml of anti-CD28 (REF: 553295; BD Pharmingen) (precoated in a rounded 96-well plate) in T$_H$0 (5 $\mu$g/ml of anti-IFN$\gamma$ [REF: 16-741185; Invitrogen], 5 $\mu$g/ml of anti-IL-12/23 [REF: 505308; BioLegend], 5 $\mu$g/ml of anti-IL-4 [REF: 554385; BD Pharmingen] and 10 ng/ml of IL-2 [404-ML-010; R&D systems]), T$_H$1 (5 $\mu$g/ml of anti-IL-4, 10 ng/ml of IL-2 and 10 ng/ml of IL-12 [419-ML-010; R&D systems]), T$_H$2 (5 $\mu$g/ml of anti-IFN- $\gamma$, 5 $\mu$g/ml of anti-IL-12/23,10 ng/ml of IL-2, and 10 ng/ml of IL-4 [402-ML-020; R&D systems]), T$_H$17 (R and D systems, CDK017, as per the manual's instructions), and iTreg (5 $\mu$g/ml of anti-IFN$\gamma$, 5 $\mu$g/ml of anti-IL-12/23, 5 $\mu$g/ml of anti-IL-4, 2 ng/ml of IL-2 and 3 ng/ml of TGF$\beta$ [7666-MB-055; R&D systems]) conditions. Naïve CD4 cells under T$_H$0, T$_H$1, and T$_H$2 conditions were activated were 48 h and were further maintained in T$_H$0/1/2 polarising conditions for another 3–5 d. Naïve CD4 cells under T$_H$17 and iTreg conditions were activated for 72 h and were further maintained in T$_H$17/Treg conditions for another 3–4 d. To confirm CD4 helper differentiation, 1 × 10$^5$ of T$_H$0, T$_H$1, T$_H$2 and were stimulated either with 2 $\mu$g/ml of anti-CD3 (plate-bound) or with 100 ng/ml of PMA (P8139; Sigma-Aldrich)

and 0.5 µg/ml of ionomycin (I0634-1MG; Sigma-Aldrich), in a rounded 96-well plate for 5 h in the presence of 2 µM of monesin (Cay16488-1; Biomol). $T_H$17 cells were stimulated as per the manufacturer's instructions (CDK017; R and D systems). After stimulation, cells were analysed for intracellular expression of IFN-γ, IL-4, and IL-17 by flow cytometry. The in vitro differentiated Tregs were analysed for the intracellular expression of FOXP3 by flow cytometry. After confirmation of CD4 helper differentiation, *Satb1* expression levels were analysed by flow cytometry, immunoblotting, and qRT-PCR. Culture supernatants of PMA and ionomycin-treated CD4 $T_H$0 and CD4 $T_H$2 cells (in the absence of monesin) were also recovered, flash-frozen in $N_2$, and stored in −80°C. The quantification of mIL-4 and mIL-5 cytokines in these supernatants was performed by the Laboratory for Immunogenomics (RIKEN, IMS) via Luminex analysis.

## Western blotting

T cells in complete medium were pelleted down by centrifugation at 200$g$ for 5 min at 4°C. Supernatants were aspirated, and the cell pellets were resuspended in 1 ml of ice-cold PBS. Cells were transferred into 1.5 ml Eppendorfs and pelleted down by centrifugation at 200$g$ for 5 min at 4°C. The supernatants were aspirated and pellets were washed in 1 ml of ice-cold PBS. The cells were pelleted down by centrifugation at 200$g$ for 5 min at 4°C and supernatants were aspirated. Cells were lysed in lysis buffer (2% SDS in 50 µM Tris–HCL pH.8, supplemented with EDTA-free protease inhibitor, Roche) at cell concentration of $20 × 10^6$/ml and incubated at 95°C for 15 min. Debris were pelleted down by centrifugation at 13,000$g$ for 10 min at room temperature and protein lysates were transferred into new 1.5 ml Eppendorf tubes. Protein lysates were adjusted with 2X laemil sample buffer (Bio-Rad) + 2-β-mercaptoethanol and boiled at 95°C for 10 min. Each lane was loaded with the equivalent of 100,000 T cells and separated by SDS–PAGE in 10% polyacrylamide gels (e-PAGEL, ATTO, EHR-T10L). Proteins were then transferred onto polyvinylidene difluoride membrane (REF:1704156; Bio-Rad) and blots were blocked with TBST (REF: 12749-21, 0.05% vol/vol; Nacalai tesque) + 5% (wt/vol) of milk for 1 h. All antibodies were used accordingly to the manufacturer`s instructions. Blots were then probed with the following primary antibodies in TBST (0.05% vol/vol) + 5% (wt/vol) of milk, overnight at 4°C: SATB1 (ab109122; Abcam), GAPDH (REF: 2275-PC-100; R and D systems) and SMC1 (ab9262; Abcam). The blots were then wash three times in 15–30 ml TBST (0.05% vol/vol) before incubation in HRP-conjugated secondary antibodies (REF:65-6120; Invitrogen) in TBST (0.05% vol/vol) + 5% (wt/vol) of milk. The blots were then wash three times in 15–30 ml TBST (0.05% vol/vol) and chemiluminescence were quantified on the Amersham Imager 680 (GE Healthcare). All immunoblots shown are representative of at least three biological replicates.

## qRT–PCR

$1 × 10^6$ of $T_H$0, $T_H$1 or $T_H$2 cells were pelleted down by centrifugation at 200$g$, at 4°C for 5 min. Supernatants were discarded and cell pellets were immediately resuspended in 750 µl of Trizol (REF: 15596026; Ambion). Samples were snap-frozen in liquid $N_2$. For RNA

**Table 1. List of cDNA primers used for qRT-PCR.**

| cDNA primer name (mouse) | Sequence |
| --- | --- |
| Satb1-F | CCCTCTAGGAAGAGGAAGGC |
| Satb1-R | GTTCCACCACGCAGAAAACTGG |
| Gm19585-F | CCCGTCTAAAGGATGTGGAATTGGA |
| Gm19585-R | GGCCATCCACTAGGAATACCCA |
| Hprt-F | GTCGTGATTAGCGATGATGAACC |
| Hprt-R | ATGACATCTCGAGCAAGTCTTTCAG |
| Gata3-F | GCAGAACCGGCCCCTTATCAA |
| Gata3-R | GTCTGACAGTTCGCGCAGGA |
| Il4-F | TCGGCATTTTGAACGAGGTC |
| Il4-R | GAAAAGCCCGAAAGAGTCTC |
| Il5- F | CCGTGGGGGTACTGTGGAAATG |
| Il5-R | TCCGTCTCTCCTCGCCACAC |
| Il13-F | GTTCTGTGTAGCCCTGGATTCCC |
| Il13-R | CCGTGGCGAAACAGTTGCTT |

isolation, cell lysates were thawed, and chloroform added (200 µl per 1 ml of Trizol). RNA samples were extracted as per the manufacturer's instruction. The samples were vortexed for 20 s and incubated at room temperature for 3 min. The samples were pelleted down by centrifugation at 12,000$g$ at 4°C for15 min. The top aqueous phase containing RNA was transferred into a new tube. RNA samples were purified by using Zymogen RNA clean and concentrator kit (R1017; Zymo research), as per the manufacturer's instructions. RNA concentrations were obtained on a NanoDrop Spectrophotometer and 1.2 µg of RNA were subjected to cDNA synthesis by using the Super VILO cDNA synthesis kit (REF:11756050; Thermo Fisher Scientific). qRT–PCR was performed using Power up SYBR green master mix (REF: A25742; applied biosystems) on the QuantStudio three Real-Time PCR Systems, accordingly to the manufacturer's instructions. The following gene expression levels analysed by qRT–PCR are listed in Table 1.

## *Gm19585* shRNA lentiviral production

The RNAi consortium was used to generate three shRNAs for the knockdown of *Gm19585* derived transcripts (see Table 2). shRNA sequences were cloned into the *PLKO.gfp* cloning vector, which was kindly gifted by Dr. Jun Huh (Harvard University): cloning, transfection, and production of lentiviral particles were performed accordingly to Addgene's protocol. After screening for shRNA inserts, lentiviruses for individual shRNAs were produced in HEK293T cells plated in six-well plates in antibiotic-free DMEM (Gibco, supplemented with glutamine and 10% FBS). For one transfection assay and using the Fugene HD transfection reagent (REF: E1312; Promega) accordingly to the manufacturer`s instructions, the HEK293T cells were cultured in 10% FBS-containing DMEM and were transfected with 1 µg of shRNA plasmid, 750 ng of psPAX2 packaging plasmid, and 250 ng of pMD2.G envelope plasmids. Packaging and envelope plasmids were obtained from Addgene. Viral supernatants were collected

**Table 2.   List of shRNA primers used for Gm19585 knockdown study.**

| shRNA primer name (mouse) | Sequence |
|---|---|
| Gm19585-ex1 shRNA-F: | CCGGTCTAAGGTCCTAGGGTCTTTACTCGAGTAAAGACCCTAGGACCTTAGATTTTTG |
| Gm19585-ex1 shRNA-R: | AATTCAAAAATCTAAGGTCCTAGGGTCTTTACTCGAGTAAAGACCCTAGGACCTTAGA |
| Gm19585-ex2 shRNA-F: | CCGGGGACTTGCTTACCCGTCTAAACTCGAGTTTAGACGGGTAAGCAAGTCCTTTTTG |
| Gm19585-ex2 shRNA-R: | AATTCAAAAAGGACTTGCTTACCCGTCTAAACTCGAGTTTAGACGGGTAAGCAAGTCC |
| Gm19585-ex4 shRNA-F: | CCGGAGCAGAGTCATTTACTTATTACTCGAGTAATAAGTAAATGACTCTGCTTTTTTG |
| Gm19585-ex4 shRNA-R: | AATTCAAAAAAGCAGAGTCATTTACTTATTACTCGAGTAATAAGTAAATGACTCTGCT |

3 d post transfection and filtered through 0.45 µm low protein-binding filters. Polybrene (Sigma-Aldrich) was added to the viral supernatants at a final concentration of 2 µg/ml, aliquoted, and stored at –80°C.

### $Gm19585$ knock-down in in vitro T$_H$2 CD4 T cells

CD4 T cells from wild-type mice were isolated from pLNs (axillary, inguinal, and cervical) by using EasySep mouse CD4 T cell isolation kit (ST-19752). Cells were stimulated with 2 µg/ml of immobilised anti-CD3 and anti-CD28 in T$_H$2 polarising conditions (5 µg/ml of anti-IFN-γ, 5 µg/ml of anti-IL-12/23,10 ng/ml of IL-2, and 10 ng/ml of IL-4) for 24 h. After activation, cells were pelleted down by centrifugation at 200$g$, at room temperature for 5 min and resuspended in 20 µl of prewarmed PBS. The cells were added to 500 µl of thawed viral supernatants and put into 48-well plates. The cells were centrifuged at 660$g$ at 32°C for 1.5 h and were rested in the tissue culture incubator for 1 h 1 volume of 2X T$_H$2 cytokines (5 µg/ml of anti-IFN-γ, 5 µg/ml of anti-IL-12/23,10 ng/ml of IL-2, and 10 ng/ml of IL-4) were then added and the cells were incubated for another 48 h $pLKO$-$Gfp$ (empty) and $pLKO$-$Gfp$-$Gm19585$ shRNA-transduced cells were monitored and T$_H$2 cytokines were refreshed every second day. CD4$^+$GFP$^+$ cells were then sorted on 6 d post transduction and immediately lysed in Trizol for qRT–PCR analyses.

### ATAC-seq analyses

ATAC-Bigwig files from Immgen database (19) were analysed and visualized on the IGV genome browser. Genomic regions of interest were then further analysed on the USCS genome browser for conservation.

### 3e HiC data analyses

3e HiC datasets from mESCs (GSM1620000) and CD4 T$_H$2 cells (GSM1619998) were downloaded from the Gene Expression Omnibus (GEO) site with the accession numbers GSE66343 (22). HiC datasets from thymocytes (GSM5268043) were downloaded from GEO site with the accession number GSE173476 (23). To examine the chromosome structure and topologically associated domain structures surrounding $Satb1$ and $Satb1$-$a$ (exon 4 of $Gm19585$) loci, reads on chromosome 17 were extracted and realigned on mouse genome mm10 using bowtie2 (version 2.4.6). Mapped reads were collected and converted into.hic format using Juicer tools (version

**Table 3.   List of primers used for 3C qRT-PCR.**

| 3C primer name (mouse) | Sequence |
|---|---|
| S1: | CTGGCAGGTTAGCAAGGCAAGCTC |
| S2: | TCAGGTGAGAGGCCTGAGCACAAC |
| AS1: | CAGGAAACATCCCTCAGGAAGATG |
| AS2: | GCAGAATGGCCCAGAGTATGAGAG |
| Gapdh F: | ACAGTCCATGCCATCACTGCC |
| Gapdh R: | GCCTGCTTCACCACCTTCTTG |

1.22.01, https://github.com/aidenlab/juicer) and visualized using Juicebox (version 1.6, https://github.com/aidenlab/Juicebox/).

### H3K27 acetylation and STAT6 ChIP-seq analyses

Publicly available H3K27Ac ChIP-seq datasets from murine CD4$^+$ FOXP3$^-$ and CD4$^+$FOXP3$^+$ T cells (8) were used to analyse H3K27Ac peaks at the $Satb1$-$a$ and $Satb1$-$b$ loci. The Sequence Read Archive (SRA) files SR5385345 and SRR5385345 were downloaded from SRA website Similarly, STAT6 ChIP-seq in CD4 T$_H$2 cells (24), was used to analyse STAT6 DNA-binding sites at the $Satb1$-$a$ locus and the SRA file SRA054075 was downloaded from SRA website. $Fastdump$ was used to convert all SRA files into fasta, which were processed for the removal of adaptor sequences and then realigned to mm10 reference genome using Bowtie2. Samtools was then used to sort and convert the SAM files (output files, generated from the genome alignment) into BAM files. BAM files were then indexed using samtools for the generation of bigwig files. The bigwig files were then visualized on the IGV genome browser.

### 3C assay

Procedures for 3C are described in our previously published article (29). The primer sequences used for the 3C assay qRT-PCR are provided in Table 3.

### $A.$ $alternata$ induced airway inflammation model

8–12 wk old female mice were anesthetized by isofluorane inhalation, followed by intra-nasal or intra-tracheal injection of $A.A.$ extract (ITEA, 10117., 20 µg per head, in 40 µl of PBS) on Day 0, Day 3, Day 6, and/or Day 9. After 24 h post final challenge, naïve and $A.A.$

treated female mice were euthanized by $CO_2$ inhalation. For cytokine analyses, BAL fluid (BALF) was first collected by intratracheal insertion of a catheter and 1 lavage of 500 µl of HBSS (2% FBS) and transferred into new 1.5 ml Eppendorf tubes. The extracted BALF was immediately pelleted down by centrifugation (300$g$, 5 min at 4°C) and BALF supernatant were transferred into new 1.5 ml Eppendorf tubes. The pelleted cells were resuspended in 500 µl of HBSS (2% FBS). BALF supernatants were flash-frozen in liquid $N_2$ and stored in –80°C. Additional two lavages of 500 µl of HBSS (2% FBS) were used to further collect the BALF of naïve and AA challenged mice and were combined with BALF cells from the first lavage, totalling to a final volume of 1.5 ml. BALF cells were kept on 4°C for flow cytometric analyses. Lung cell suspensions from mice were prepared as outlined above. After Percoll gradient centrifugation, CD4 $T_H$2 cells (CD45$^+$Thy1.2$^+$CD4$^+$ST2$^+$) and ILC2s (CD45$^+$Thy1.2$^+$CD4$^-$ST2$^+$) were sorted using FACS Aria. 10,000 CD4 $T_H$2 cells and ILC2s were activated with PMA (100 ng/ml) and ionomycin (0.5 µg/ml) in a rounded 96-well plate for 4 h in the presence of Monesin and were then subjected to flow cytometry analyses for intracellular cytokines levels. The quantification of the mIL-5 cytokine in BAL supernatants of naïve and *A.A.*-treated mice were also performed by the Laboratory for Immunogenomics (RIKEN, IMS) via Luminex analysis.

### OVA-induced airway inflammation model

8–12-wk-old male and female mice were first immunized with intraperitoneal administration of 50 µg of OVA (in 100 µl PBS) + 100 µl of Alum (Imject Alum Adjuvant, Thermo Fisher Scientific), on Day 0 and Day 13. On days 27, 28, and 29, the mice were anesthetized by isoflurane inhalation, followed by intranasal injection of 50 µg of OVA (in 40 µl PBS). After 24 h post final challenge, BAL and lungs of naïve and OVA-challenged mice were processed and analysed using the same methods for *A.A*-induced airway inflammation model.

### In vitro activation of ILC2s

Lung cell suspensions from mice were prepared as outlined above. After Percoll gradient centrifugation, 5,000 ILC2s (CD45$^+$Lin$^-$CD3$\varepsilon^-$Thy1.2$^+$) were sorted into each well of 96 well-plate. Sorted ILC2s were cultured in 100 µl of complete medium (KOHJIN BIO DMEM-H [16003016] + 10% FBS) in the presence of mIL-33 (10 ng/ml), mIL-2 (10 ng/ml) and mIL-7 (10 ng/ml) for 4 d.

### Statistical analysis

Student T tests and two-way ANOVA with post hoc Bonferroni tests were performed using GraphPad Prism software: \*\*\*$P < 0.001$, \*\*$P < 0.01$, and \*$P < 0.05$.

## Supplementary Information

## Acknowledgements

We thank Noriko Yoza for her continuous help with cell sorting and Yusuke Iizuka and RIKEN IMS Animal group for ES aggregation and genome editing. We`d also like express our gratitude to Chizuko Miyamoto, Yuria Taniguchi, Miho Mochizuki, and Natsuki Takeno for their assistance with experiments and genotyping. This work was supported by the MITSUBISHI FOUNDATION (201910029) and Ministry of Education, Culture, Sports, Science and Technology Grants-in-Aid for Scientific Research on Innovative Areas "Replication of Non-Genomic Codes" (JP19H05747) (I Taniuchi) and by support funding from RIKEN's Gender Equality Program (A Nomura).

### Author Contributions

A Nomura: formal analysis, investigation, and writing—original draft, review, and editing.
T Kobayashi: formal analysis.
W Seo: formal analysis.
M Ohno-Oishi: formal analysis.
K Kakugawa: formal analysis.
S Muroi: methodology.
H Yoshida: resources, formal analysis, and methodology.
TA Endo: resources, software, and formal analysis.
K Moro: formal analysis.
I Taniuchi: conceptualization, supervision, and writing—original draft, review, and editing.

### Conflict of Interest Statement

The authors declare that they have no conflict of interest.

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
