## [Reviewer comments · Life Science Alliance]

Life Science Alliance

Identification of a novel enhancer essential for Satb1 expression in TH2 cells and activated ILC2s.

Aneela Nomura, Tetsuro Kobayashi, Wooseok Seo, Michiko Ohno-Oishi, Kiyokazu Kakugawa, Sawako Muroi, Hideyuki Yoshida, Takaho Endo, kazuyo Moro, and Ichiro Taniuchi

DOI: <https://doi.org/10.26508/lsa.202301897>

Corresponding author(s): *Ichiro Taniuchi, RIKEN Center for Integrative Medical Sciences*

Review Timeline:

Submission Date:	2023-01-03
Editorial Decision:	2023-02-02
Revision Received:	2023-04-11
Editorial Decision:	2023-05-05
Revision Received:	2023-05-08
Accepted:	2023-05-08

Scientific Editor: Novella Guidi

Transaction Report:

February 2, 2023

Re: Life Science Alliance manuscript #LSA-2023-01897-T

Dr. Ichiro Taniuchi
RIKEN, Center for Integrative Medical Sciences (IMS)
RIKEN Center for Integrative Medical Sciences
1-7-22, Suehiro-cho, Turumi-ku
Turumi-ku
Yokohama, Kanagawa 230-0045
Japan

Dear Dr. Taniuchi,

Thank you for submitting your manuscript entitled "Identification of a novel enhancer essential for Satb1 expression in TH2 cells and activated ILC2s." to Life Science Alliance. The manuscript was assessed by expert reviewers, whose comments are appended to this letter. We invite you to submit a revised manuscript addressing the Reviewer comments.

Thank you for this interesting contribution to Life Science Alliance. We are looking forward to receiving your revised manuscript.

Sincerely,

B. MANUSCRIPT ORGANIZATION AND FORMATTING:

Reviewer #1 (Comments to the Authors (Required)):

In this paper, Nomura et al exploited a SATB1-Venus reporter mouse model and precisely evaluated the SATB1 expression levels of T-lineage cells. By using genome editing methodology, the authors have determined one cis-regulatory enhancer for Satb1 expression in CD4 TH2 cells. Then, the authors have proved that this enhancer, named as Satb1-Eth2 (enhancer for TH2 cells) is indispensable to up-regulate and maintain Satb1 expression in CD4 TH2 cells and group 2 innate lymphoid cells (ILC2s). Information from public data base suggested the interaction of Stat6 with this enhancer. The results of in vitro experiments have clearly shown that Satb1-Eth2 is essential for repressing IL-5 expression in CD4 TH2 cells though it remains elusive how this enhancer and Satb1 are coordinately involved in physiological regulation of IL5 and allergic reaction in vivo.

Although this paper includes quite a few negative data, all the experiments seem to have been well conducted and precisely executed. Those negative results also have some values. This reviewer has no major concern on this meticulous study, but wants to make a couple of minor comments.

1. Page 6 line124-127

Notably, DP thymocytes undergoing positive selection (CD69+TCR β mid and CD69+TCR β hi) showed further increase in SATB1-Venus expression (Figure 1B), indicating that SATB1 expression is positively controlled by TCR stimulation.

---- This is perhaps an overstatement because this paper provides no data directly linking TCR stimulation and SATB1 expression.

2. The text includes a few grammatical mistakes.

Reviewer #2 (Comments to the Authors (Required)):

In this paper, these authors study how SATB1 expression is regulated. The authors identify an enhancer for Satb1 expression in Th2 cells and ILC2. While the deletion of this enhancer decreases the Satb1 expression, the authors not find functional relevancy of SATB1 decrease in Th2 cells or ILC2, in vivo. Only an increase in IL5 expression in vitro differentiated Th2 cells are observed.

In the abstract, the authors do not describe about all the conclusions they can draw from their results. They only talk about the results in favour of their title.

For example, they do not say that the enhancer deletion has no functional relevance in vivo.

Throughout the article (in particular in introduction), they talk about Satb1 regulates/controls without saying in which sense activation/inhibition, expression/repression. It is difficult for the reader to follow the different studies.

The authors characterized the Satb1-venus fusion reporter mouse in thymus and periphery. The authors describe the venus expression by cytometry by analyzing the MFI only. the authors do not report the % of cells that came positive, only the decrease of venus expression on all population (positive + negative). For example in Figure 1E only a part of effector memory cells are positive for Venus. Same in Figure EV1F. Why are these cells divided into Venus positive and negative cells? What is the functional relevance?

What is the functional relevance of these different levels of Venus (Satb1 expression) in thymus and in periphery ?

The authors conclude that T cell activation in the periphery induces down-regulation of Satb1 expression. It is not a down regulation in all the cells, but the lost in one population of cells. Why ? There is a correlation with proliferative/ activated cells ? More experiments with a kinetic on TCR-activated naïve CD4 cells are necessary.

The authors conclude that SATB1-venus could serve as a good marker for separating effector and naïve cells. However, the authors do not show that all effector cells are negatives for Venus and naïve cells are positives for venus.

The authors indicate that Satb1 expression declines after activation of T cells but they do not show the expression of Satb1 in naïve CD4 T cells before the Th differentiation (Figure 1G).

Line 169: Figure S2A ??? The authors do not show the correlation between Venus expression and SATB1 mRNA level.

Figure 2B line 202 "significantly reduced": MFI 2573 to 2044; What is the functional relevance ? what is the % of cells that express venus ? Why the deletion of enhancer do not decrease the expression in all the cells ?

Figure 2C: It is important to increase the number of experiments to do statistical.

Figure EV2D: the quantification of the western blot is necessary.

Using public data, the authors show that Sabt1-A region as a genomic enhancer via STAT6. The authors need to demonstrate this mechanism, by performing an ChIP anti stat6 in wt and deleted Sabt1-a region mice.

In Satb1 deltaEth2/ deltaEth2, the authors do not analyse the development of T cells in periphery, while the enhancer is important in effector CD4 T cells.

The authors need to analyse the IL4, IL5 and IL13 production after PMA/iono, but also after TCR engagement.

In vivo, using Alternaria Alternate model, the authors show no physiological relevance of the absence of the Sabt1. In others models as asthma (house dust mite or ovalbumin models) ???

In general manner, in different figures the statistics are no present.

The paper need a functional relevancy.

Reviewer #3 (Comments to the Authors (Required)):

The authors constructed a SATB1-Venus expressing reporter mouse and investigated SATB1 expression in various stages of T cell development and multiple subpopulations of T cells. Using this reporter system, the authors investigated the roles of two candidates of distal enhancers in SATB1 expression regulation in T cells. The authors found that the Satb1-a, one of these two sequences can regulate the expression of SATB1 in Th2 and ILC2 cells. The authors further analyzed the mechanism of regulation of SATB1 expression by this regulatory sequence in Th2 cells. The authors also analyzed the effect of this regulatory sequence on the Th2 immune response. The study contributes to the understanding of the complex regulatory mechanism of SATB1 in T cells. But there are still some concerns with the current manuscript.

Major concerns:

1. This study needs to be very cautious about the extent to which the SATB1-Venus reporter system constructed by the author can truly reflect the protein expression level of SATB1. Indumathi Patta et al. used Western blot to observe that the level of SATB1 protein was higher in CD4 SP cells than in DP cells (Nucleic Acids Research, 2020, Vol. 48, No. 11 5873-5890), which is inconsistent with the observation here using SATB1-Venus. Also, I noticed that the Th1 MFI of Satb1+/Venus mice was higher than Th2 (Fig 2C), but Western Blot showed that SATB1 protein was higher in Th2 than in Th1 (Figure EV2D). This inconsistency made me suspect that SATB1-Venus could not truly reflect SATB1 protein levels. Therefore, the author needs to use Western Blot or other alternative methods to confirm that the protein level expression of SATB1 is consistent with that of SATB1-Venus.

2. Fig EV1E. The altered ratio of CD8SP in Satb1 Venus/Venus mice is evident and it can not be ignored. The authors need to explain this in the manuscript.

3. On page 7, line 150, the author says "The SATB1-Venus hi IEL population expressed CD62L, with their CD4 and CD8alpha expression profiles similar to that in splenic T cells". The authors should provide data and analysis.

4. The authors used ATAC-seq data to search enhancers, which is somewhat limited. CTCF binding sites and silencers are also accessible regions. While the histone modification H3K27 acetylation is a better enhancer marker, it would be better to combine the ChIP-seq data of H3K27 acetylation to analyze the candidate enhancers of SATB1. ChIP-seq of H3K27 acetylation can be added to Fig 2A to help understand enhancer characteristics.

5. The resolution of the Fig 3B Hi-C heatmap is too low, and the black line is too thick, obscuring the details of the SATB1 locus in the heatmap. Suggest a better presentation of the Hi-C data. In addition, the heat map of Hi-C could not provide the specific interaction between Satb1-a and the SATB1 promoter. It is recommended to perform a 4C assay or similar techniques to confirm the interaction between the enhancer and the promoter.

6. The mice information in this manuscript is confusing. What is the difference between SATB1 Venus-Δa, SATB1 ΔEth2, and SATB1 Venus-ΔEth2? The authors should state this clearly in the manuscript. Since the CRISPR-Cas9 system excision is not perfect, there will be some variation in deletion regions. What is the difference between the deletion sequences in these mice? The authors should provide sequencing validation results for each mouse line.

Minor:

1. Cd4-cre (Line114, Page5) or CD4-cre (Figure EV1E)? Need to be consistent.

2. Figure 1C, 1D, 1E, and 1G have missing x-axis numbers.

3. Fig1F the conditions for FACS plots are not mentioned. The same issues are applicable to all FACS plots.
4. Page 7 Line 169: Figure "S2A" should be "EV2A".
5. Fig 4A right panel "Satb1 Δ Eth2 Δ Eth2" should be "Satb1 Δ Eth2/ Δ Eth2". The same issues are also applicable to Fig EV3A.
6. Page 12 "Satb1 Δ eth2/ Δ eth2" should be "Satb1 Δ Eth2/ Δ Eth2".

Reviewer #1 (Comments to the Authors (Required)):

In this paper, Nomura et al exploited a SATB1-Venus reporter mouse model and precisely evaluated the SATB1 expression levels of T-lineage cells. By using genome editing methodology, the authors have determined one cis-regulatory enhancer for Satb1 expression in CD4 TH2 cells. Then, the authors have proved that this enhancer, named as Satb1-Eth2 (enhancer for TH2 cells) is indispensable to up-regulate and maintain Satb1 expression in CD4 TH2 cells and group 2 innate lymphoid cells (ILC2s). Information from public data base suggested the interaction of Stat6 with this enhancer. The results of in vitro experiments have clearly shown that Satb1-Eth2 is essential for repressing IL-5 expression in CD4 TH2 cells though it remains elusive how this enhancer and Satb1 are coordinately involved in physiological regulation of IL5 and allergic reaction in vivo.

Although this paper includes quite a few negative data, all the experiments seem to have been well conducted and precisely executed. Those negative results also have some values. This reviewer has no major concern on this meticulous study but wants to make a couple of minor comments.

We are very thankful for Reviewer #1`s positive evaluation on our study.

1. Page 6 line124-127

Notably, DP thymocytes undergoing positive selection (CD69+TCR β mid and CD69+TCR β hi) showed further increase in SATB1-Venus expression (Figure 1B), indicating that SATB1 expression is positively controlled by TCR stimulation.

---- This is perhaps an overstatement because this paper provides no data directly linking TCR stimulation and SATB1 expression.

We thank Reviewer #1 for this suggestion. We agree that we did not provide any evidence showing a direct link of TCR stimulation with SATB1 upregulation in thymocytes. Therefore, we rewrote this statement to point out this possibility in the revised manuscript below in line 128, on page 6.

“which may suggest that *Satb1* expression is positively regulated during positive selection signals.”

2. The text includes a few grammatical mistakes.

We appreciate errors pointed out by this reviewer. We have now corrected the grammatical errors as much as possible.

Reviewer #2 (Comments to the Authors (Required)):

In this paper, these authors study how SATB1 expression is regulated. The authors identify an enhancer for Satb1 expression in Th2 cells and ILC2. While the deletion of this enhancer decreases the Satb1 expression, the authors not find functional relevancy of SATB1 decrease in Th2 cells or ILC2, in vivo. Only an increase in IL5 expression in vitro differentiated Th2 cells are observed.

In the abstract, the authors do not describe about all the conclusions they can draw from their results. They only talk about the results in favour of their title.

For example, they do not say that the enhancer deletion has no functional relevance in vivo.

First, we thank Reviewer #2 for his/her constructive comments. In the title and abstract, we wanted to avoid overstating our results: we mainly focused on describing the key findings derived from our study. In terms of “functional relevance of enhancer deletion in vivo”, we have confirmed that, in the response to lung inflammation, the Satb1-Eth2 enhancer is required to maintain SATB1 expression in both *in vivo* Th2 cells and activated ILC2s. Thus, we believe that this enhancer, which we identified, has a functional relevance in regulating SATB1 expression *in vivo*. The problem is that we have yet to understand how this enhancer and SATB1 are co-ordinately involved in physiological regulation of IL5 to finely regulate allergic reaction *in vivo*. We are afraid that the Reviewer #2 is possibly misdirected by the results that showed no significant effects of enhancer deletion in the Th2 immune responses. As we pointed out in the discussion section, these negative results do not exclude the possible roles of this enhancer/low Satb1 expression in Th2/ILC2 in regulating Th2 immune responses under other experimental settings, which again would require additional intensive studies. Therefore, at this moment, we were afraid to make a strong statement on this point in the abstract to prevent misleading to our readers.

Throughout the article (in particular in introduction), they talk about Satb1 regulates/controls without saying in which sense activation/inhibition, expression/repression. It is difficult for the reader to follow the different studies.

We thank the Reviewer #2 for this comment. We have revised the text, aiming to increase the clarity and for easy reading.

The authors characterized the Satb1-venus fusion reporter mouse in thymus and periphery. The

authors describe the Venus expression by cytometry by analyzing the MFI only. the authors do not report the % of cells that came positive, only the decrease of Venus expression on all population (positive + negative). For example in Figure 1E only a part of effector memory cells are positive for Venus. Same in Figure EV1F. Why are these cells divided into Venus positive and negative cells? What is the functional relevance? What is the functional relevance of these different levels of Venus (Satb1 expression) in thymus and in periphery?

We thank Reviewer #2 for raising this important question. First, and as requested, we have provided the frequency of SATB1-hi and SATB1-lo populations within naïve/central/effector memory T cell populations in Fig. S11.

We had the same question raised by Reviewer #2 why there are Satb1-high and -low cells in CD44⁺CD62L⁻ CD4 T cells subsets. We wished to address this but could not find good experimental approaches to directly answer this question. If Reviewer #2 can suggest us specific experiments that would address this question, we are willing to perform them.

As such, at this moment, we did not reveal the functional relevance of these heterogenous SATB1-Venus levels within these central and effector memory T cell populations. However, we would ask the Reviewer #2 to appreciate the merit of the *Satb1-Venus* reporter model. Without this fluorescent reporter model, it would be impossible to quantify SATB1 protein level in various T cell subpopulations with a single cell resolution. Thus, heterogenous expression of Satb1 protein was clearly shown at the first time by our approach. In addition, to understand how *Satb1* expression is regulated in various T cell subpopulations, we utilized the *Satb1-Venus* reporter mice to validate the physiological function of *cis*-regulatory candidate regions, and eventually we have identified a novel Th2 specific enhancer regulating *Satb1* expression not only *in vitro* but also *in vivo* differentiated Th2 cells. We believe that our approach is one of the most reliable ways to address the functional relevance of the different levels of SATB1 by perturbing its physiological regulation. Of course, the *Satb1-Eth2* enhancer is not the sole regulatory region for *Satb1* expression and comprehensive understanding of the functional relevance of heterogenous SATB1 expression will require additional genetic approaches and intensive characterization of these SATB1-hi and SATB1-lo central/effector memory T cell populations, which we believe is beyond the scope of this manuscript.

The authors conclude that T cell activation in the periphery induces down-regulation of Satb1 expression. It is not a down regulation in all the cells, but the lost in one population of cells. Why ? There is a correlation with proliferative/ activated cells ? More experiments with a kinetic on TCR-activated naïve CD4 cells are necessary.

We thank Reviewer #2 for raising these points and apologise for not showing frequencies of SATB1-Venus^{Hi} and SATB1-Venus^L cells within naïve, central memory and effector memory T cell subsets. We now provide this data in Figure S1I. As addressed in our previous comment and in our FACS analyses of *ex vivo* cells, naïve T cells showed a uniform and high SATB1-venus expression, whereas in other T cell subsets, such as CD44⁺ population, contains cells showing low SATB1-venus expression. Since this is an *ex vivo* analyses of T cells, it is impossible to analyse kinetics of T cell activation in terms of when and where they are stimulated/activated.

Regardless, we also monitored SATB1-Venus expression after TCR activation (α -CD3 and α -CD28 antibodies) of naïve CD4 T cells after 24, 48 and 72 hours and have now provided the results in Fig 1G of the revised manuscript. In this experimental setting, SATB1 expression is increased after 24 and 48 hours of T cell activation and plateaued at 72 hours. According to these results obtained from *in vitro* stimulations, we revised our statement in line 165-173 on page 7.

The authors conclude that SATB1-venus could serve as a good marker for separating effector and naïve cells. However, the authors do not show that all effector cells are negatives for Venus and naïve cells are positives for venus.

The authors indicate that Satb1 expression declines after activation of T cells, but they do not show the expression of Satb1 in naïve CD4 T cells before the Th differentiation (Figure 1G).

We thank Reviewer #2 for raising these critical comments, which are related to above criticisms by the same reviewer. We agree that we did not examine all effector cells differentiated *in vivo* for Satb1-expression, but this is practically very difficult since in general not all effector cells have not been identified. But our data confirmed that all naïve T cell shows uniform and high SATB1-venus expression pattern. Thus, our statement was based on the different SATB11-Venus expression levels in different T cell populations developed *in vivo*.

As requested, we have repeated the CD4 T-helper cultures to include Satb1-Venus expression before differentiation (sorted naïve CD4 T cells prior to activation/differentiation) and 5 days after Th0/1/2/17iTreg differentiation, all on the same flow cytometric settings, and present the data in Fig 1H of the revised manuscript. These Th cell subsets differentiated *in vitro* retain Satb1-venus expression with different levels. Again, having this discrepancy between *in vivo* primed and *in vitro* stimulated cells, we toned down our statement and revised the text in lines 141-162 on pages 6-7.

Line 169: Figure S2A ???? The authors do not show the correlation between Venus expression and SBT1 mRNA level.

We thank Reviewer #2 for pointing out this mistake. We now modified the figures to show side-by-side comparisons of SATB1-Venus versus *Satb1* mRNA levels from the IMMGEN data base, in Figure 1B and 1D.

Figure 2B line 202 "significantly reduced": MFI 2573 to 2044; What is the functional relevance ? what is the % of cells that express venus ? Why the deletion of enhancer do not decrease the expression in all the cells ?

We thank the Reviewer #2 for raising this question. As we show in Figures 1E and S1I and when compared to non-venus control (wildtype) cells, all (100%) naïve CD4 T cells expressed SATB1-Venus (i.e. they are all SATB1-Venus hi). In Figure 2B, deletion of the *Satb1-a* enhancer caused a small but significant decrease in SATB1-Venus MFI (this is clearly judged by the shift of SATB1-Venus peak, toward the left-hand side). Thus, the deletion of the *Satb1-a* enhancer promoted a small decreased in SATB1-Venus expression in all naïve CD4 T cells. We did elaborate on this finding in our discussion on why would there a small decline in SATB1-Venus expression in naïve CD4 T cells (see lines 422-434, page 17).

As mentioned throughout our manuscript, there are STAT-DNA binding sites in the *Satb1-a* regulatory region, which is indeed occupied by STAT6, but can also be accessible for other STAT members, such as STAT5. Naïve T cells require the IL-7 cytokine for their homeostatic proliferation, and therefore, it is possible that *Satb1-a* is also occupied by STAT5, to maintain SATB1 expression. But since we only observed a small decrease in SATB1 expression in *Satb1-Δa* mice in naïve CD4 T cells, it is, thus, likely that other prominent regulatory regions are highly involved in regulating SATB1 expression in naïve T cells, which are unfortunately yet to be identified. Similarly, there would be other prominent regulatory regions in regulating SATB1 expression in other types of cells. This could be a reason why the deletion of one enhancer did not decrease the expression in all the cells. We are willing to extend our reply to the discussion section if the Reviewer #2 requests it.

Figure 2C: It is important to increase the number of experiments to do statistical

We thank Reviewer #2 for raising this point. We have now added additional biological replicates from *Satb1*^{+Venus} and *Satb1*^{+Venus-Δa} mice in Figure 2C to support our conclusion from the statistical viewpoint.

On the other hand, please note that we could not add additional replicates from *Satb1*^{+Venus-Δb} mice, as we have not been breeding them since Sept 2022 after we consistently observed no significant differences in SATB1-Venus expression in *in vivo* T cell subpopulations in the

Satb1^{+/*Venus-Δb*} mice as shown in Fig. 2B and two times experiments for *in vitro* T cell differentiation. Although it was possible to retrieve this mouse strain from the frozen stock, we must consider the strict polices for conducting animal research (3Rs) and judged that additional experiments with this mouse strain would not be essential for this work. We therefore ask this reviewer and the editor to evaluate whether additional replicates from *Satb1*^{+/*Venus-Δb*} mice are necessary for publishing.

Figure EV2D: the quantification of the western blot is necessary

We thank Reviewer #2 for raising this point. We have now quantified our SATB1 immunoblots in Figure S2D and included a summarized graph with the appropriate statistical analyses.

Using public data, the authors show that Sabt1-A region as a genomic enhancer via STAT6. The authors need to demonstrate this mechanism, by performing an ChIP anti stat6 in wt and deleted Sabt1-a region mice.

We thank the Reviewer #2 for raising this point. We did attempt to provide STAT6 ChIP-qPCR result to confirm STAT6 binding in our *in vitro* differentiated CD4 Th2 cells during preparation of the original manuscript and we used 2 commercial anti-STAT6 murine antibodies.

Unfortunately, our evaluation of STAT6 ChIP experiment using *Gata3* locus (that contains known STAT6 binding region in Th2 cells) indicated that the 2 anti-STAT6 antibodies we used in ChIP did not work well probably due to low quality of antibody. Therefore, at this moment in time, it was very difficult to practically perform and optimise STAT6 ChIP-qPCR in Th0 and TH2 cells. Thus, we used the published STAT6 ChIP-seq data from murine Th2 cells to demonstrate STAT6 binding to our *Satb1-a* enhancer in the original manuscript.

In addition, please note that performing STAT6-ChIP in *Satb1-Δa* CD4 Th2 cells is practically impossible, because the STAT6 binding site in *Satb1-a* region has been deleted in *Satb1-Δa* mice.

In Satb1 deltaEth2/ deltaEth2, the authors do not analyse the development of T cells in periphery, while the enhancer is important in effector CD4 T cells

We thank the Reviewer #2 for asking this question. We found that T cell development in *Satb1* ^{Δ Eth2/ Δ Eth2} mice are normal in a steady state. We show these results for Reviewer #2 in Figure R1.

In addition, we showed in Figure 5C of our original paper that, after *Alternaria* induced lung inflammation, CD4 Th2 differentiation was unaffected in *Satb1*^{Δ*Eth2*/Δ*Eth2*} mice, suggesting that the *Satb1*-*Eth2* is dispensable for CD4 Th2 differentiation.

Figure R1: Analyses T cell development in *Satb1*^{+/+} and *Satb1*^{Δ*Eth2*/Δ*Eth2*} mice.

The authors need to analyse the IL4, IL5 and IL13 production after PMA/iono, but also after TCR engagement.

We thank the Reviewer #2 for this suggestion. We measured intracellular IL4 and IL5 production after anti-CD3 treatment and obtained similar results after PMA/iono stimulation. We present this result in Figure R2 for Reviewer #2 and have included this data in Figure S3C.

Since we didn't not see any significant differences in *Il-13* expression (Figure S3C) from our PMA/iono treated Th2 cells, we felt that that it was unnecessary to reanalyse *Il-13* expression by qPCR in our anti-CD3 treated Th2 cells.

Figure R2: Analyses of Intracellular IL-4 and IL-5 protein expression upon anti-CD3 treatment. Naïve CD4 T cells from *Satb1^{+/+}* and *Satb1 Δ Eth2/ Δ Eth2* mice were activated *in vitro*, in T_H0 or T_H2 polarizing conditions for 7 days. On day7, TH0 or Th2 cells were reactivated in anti-CD3 in the presence of Monesin for 5 hours prior to flow cytometry analyses. Data is a representative of 2 independent experiments.

In vivo, using *Alternaria Alternate* model, the authors show no physiological relevance of the absence of the *Satb1*. In others models as asthma (house dust mite or ovalbumin models) ???

We thank the Reviewer #2 for this thoughtful suggestion. We have also used the OVA-Th2 model to study the function of the *Satb1*-*Eth2*. As expected, the deletion of this enhancer did not cause any significant changes in BAL Eosinophils numbers, CD4 Th2 numbers, ILC2 numbers and IL-5 secretion (Figure R3). We have also included these results in Figures S4D and S4E.

We agree that our two *in vivo* models could not find differences in Th2 responses and IL5 expression. However, we believe that these results do not formally exclude the involvement of *SATB1* in controlling T_H2 immune responses *in vivo* under different experimental settings. Unfortunately, testing several *in vivo* models with different settings/conditions will take time. Using lung inflammation models, we clearly showed that the enhancer is also functional in activated ILC2s, which is an important and a novel finding. As suggested by the Reviewer #1, we too think that presenting and identifying this novel functional enhancer has some value, even without its detailed role in regulating Th2 responses at this point. We hope that Reviewer #2 understands our thoughts on this point.

Figure R3: Ovalbumin (OVA)-induced TH2 immune response in *Satb1*^{+/+} and *Satb1*^{ΔEth2/ΔEth2} mice. **A**, schematic shows the experimental plan for OVA-induced TH2 immune responses. *Satb1*^{+/+} and *Satb1*^{ΔEth2/ΔEth2} mice were first sensitized to OVA on Days 0 and 13 via intraperitoneal injection and then challenged with OVA on Days 25, 26 and 27 via intra-nasal injection. Lung-derived immune cells were all analyzed 24 hours post final challenge **B**, Top set of graphs show quantification of Eosinophils, CD4 TH2 cells, ILC2s and IL-5 from BAL of OVA-treated mice. The bottom set of graphs show quantification of CD4 TH2 cells, ILC2s from lungs of OVA-treated mice.

*In general manner, in different figures the statistics are no present.
The paper need a functional relevancy.*

We thank the Reviewer #2 for pointing out our errors. We have checked all figures for missing statistics.

Reviewer #3 (Comments to the Authors (Required)):

The authors constructed a SATB1-Venus expressing reporter mouse and investigated SATB1 expression in various stages of T cell development and multiple subpopulations of T cells. Using this reporter system, the authors investigated the roles of two candidates of distal enhancers in SATB1 expression regulation in T cells. The authors found that the Satb1-a, one of these two sequences can regulate the expression of SATB1 in Th2 and ILC2 cells. The authors further analyzed the mechanism of regulation of SATB1 expression by this regulatory sequence in Th2 cells. The authors also analyzed the effect of this regulatory sequence on the Th2 immune response. The study contributes to the understanding of the complex regulatory mechanism of SATB1 in T cells. But there are still some concerns with the current manuscript.

We appreciate Reviewer #3 for his/her positive evaluation on our study.

Major concerns:

1. This study needs to be very cautious about the extent to which the SATB1-Venus reporter system constructed by the author can truly reflect the protein expression level of SATB1. Indumathi Patta *et al.* used Western blot to observe that the level of SATB1 protein was higher in CD4 SP cells than in DP cells (Nucleic Acids Research, 2020, Vol. 48, No. 11 5873-5890), which is inconsistent with the observation here using SATB1-Venus. Also, I noticed that the Th1 MFI of *Satb1*^{+/Venus} mice was higher than Th2 (Fig 2C), but Western Blot showed that SATB1 protein was higher in Th2 than in Th1 (Figure EV2D). This inconsistency made me suspect that SATB1-Venus could not truly reflect SATB1 protein levels. Therefore, the author needs to use Western Blot or other alternative methods to confirm that the protein level expression of SATB1 is consistent with that of SATB1-Venus.

We thank Reviewer #3 for raising this important question.

Firstly, we think there are some technical differences in how we analysed SATB1 expression in CD4 SPs versus those in Patta *et al.* We noted that the SATB1 immunoblot data by Patta *et al.* (Figure 1G) did not specify whether the CD4 SP thymocytes were gated from CD24^{lo}TCRβ^{hi} thymocytes: it is possible that there is contamination of protein lysates coming from immature CD69⁺CD24⁺CD4⁺SP populations, which express higher SATB-1 levels than mature CD4 SP thymocytes. In our study, we gated for CD24^{lo}TCRβ^{hi}CD4⁺CD8⁻ thymocytes (see Figure S1G of our manuscript for gating strategy and *Satb1*-Venus expression) to quantify SATB1-Venus levels. These gating strategies are also very similar to those used by the IMMGEN team and they too have demonstrated that DPs express higher levels of *Satb1*-mRNA than mature CD4 SP thymocytes. We have now included the IMMGEN data in Figure 1B.

Secondly, we should caution Reviewer #3 about Dr Sanjeev Galande's published work. There have been some occasions where he was publicly criticised for mispresenting immunoblot data in at least 2 published papers (Notani *et al* 2010, PLOS Biology and Bischof, Galande *et al.* 2001, JCB (now retracted). This ultimately caused the PLOS Biology editors to express concerns about the authenticity of his data and they have now published a "Expression of concern" letter, to caution readers about the study. Moreover, JBC have withdrawn the 2001 paper, due to extreme concerns with immunoblot duplication and the overall authenticity of the data.

Nevertheless, we performed immunoblot analyses from DP and CD4 SP thymocytes from WT and *Satb1*-Venus mice (Figure R4) and found that no significant differences between endogenous SATB1 and SATB1-Venus protein in both sorted DP and CD4-SP thymocytes. Similarly, by using *Satb1*^{+/Venus} mice we showed that there was no significant differences

between endogenous SATB1 and SATB1-Venus protein in total thymocytes (Figure S1C). Therefore, we believe that SATB1-Venus does functionally represent endogenous SATB1 protein levels.

Figure R4: Analysis of SATB1 and SATB1-Venus protein expression from DP and CD4 SP thymocytes. DP (CD4⁺CD8 α ⁺) and CD4 SP (CD4⁺CD8 α ⁻) thymocytes were sorted from *Satb1*^{+/+} and *Satb1*^{Venus/Venus}. Sorted cells were resuspended in lysis buffer (5x10⁶ cells/100 μ L lysis buffer) for immunoblot analyses for SATB1. SMC1 was used as a loading control.

We also thank Reviewer #3 for pointing out the inconsistencies in our SATB1-Venus analyses in Th2 versus Th1 cell. We again performed immunoblot analyses of *in vitro* differentiated Th cell subsets, including Th17 and Treg from Wild-type and *Satb1*-Venus mice, and observed that Th1, Th2 and Th17 expressed comparably higher levels of SATB1-Venus than Th0 and Tregs (Figure R5). Our revised flow cytometry data are now included in Figure 1H. These data also show a good correlation of *Satb1*-venus intensity and endogenous *Satb1* protein levels. Having these new results, we re-checked our previous data and realized that the original data showing a higher SATB1 expression in Th2 than in Th1 shown in Fig S2D was not a representative one, therefore we replaced the old image with more representative image and eliminated exceptional data points from our analyses. We believe that our new analyses preclude concerns raised by Reviewer #3 and support our claim that SATB1-Venus does functionally represent endogenous SATB1 protein levels.

Figure R5: Analysis of SATB1-Venus protein expression in CD4 T-helper (Th) cells. Naïve CD4 T cells were sorted from *Satb1^{+/+}* and *Satb1^{+/-Venus}* mice and were polarized in T_{H0}, T_{H1}, T_{H2}, T_{H17} and iT_{Reg} for 5 days *in vitro*. **A**, Histograms show SATB1-Venus quantification in CD4 naïve versus T_{H0}, T_{H1}, T_{H2}, T_{H17} and iT_{Reg} cells from *Satb1^{+/-Venus}* mice. Graph on the right summarizes SATB1-Venus quantification from 3 independent experiments. **B**, immunoblots show expression of endogenous SATB1 proteins from T_{H0}, T_{H1}, T_{H2}, T_{H17} and iT_{Reg} cells from *Satb1^{+/+}* mice. Cells were resuspended in lysis buffer (2x10⁶ cells/100μL lysis buffer) for immunoblot analyses for SATB1. SMC1 was used as a loading control. Graph on the right summarizes endogenous SATB1 protein quantification, from 2 independent experiments.

2. Fig EV1E. The altered ratio of CD8SP in *Satb1 Venus/Venus* mice is evident and it can not be ignored. The authors need to explain this in the manuscript

We thank the Reviewer #3 for raising this important question. We agree that the ratio of CD4SP to CD8SP thymocytes appears to be increased, but further analyses of thymocyte numbers revealed no significant alterations in CD4 SP and CD8 SP numbers our *Satb1 Venus/Venus* mice. We have now adjusted Fig S1E to include CD4 SP and CD8 SP numbers and clarified this normal phenotype in the main text.

3. On page 7, line 150, the author says "The SATB1-Venus hi IEL population expressed CD62L, with their CD4 and CD8alpha expression profiles similar to that in splenic T cells". The authors should provide data and analysis.

We thank the Reviewer #3 for raising this point, we added data from both splenic and IEL derived to show CD4/CD8 expression patterns from CD62L^{hi} versus CD62L^{lo} gated T cells the data in Fig 1H in the revised manuscript.

4. The authors used ATAC-seq data to search enhancers, which is somewhat limited. CTCF binding sites and silencers are also accessible regions. While the histone modification H3K27 acetylation is a better enhancer marker, it would be better to combine the ChIP-seq data of H3K27 acetylation to analyze the candidate enhancers of SATB1. ChIP-seq of H3K27 acetylation can be added to Fig 2A to help understand enhancer characteristics

We thank the Reviewer #3 for this thoughtful suggestion. We have now included publicly available H3K27 acetylation ChIP-seq data in Fig. 2B of the revised manuscript. This dataset is derived from CD4 FOXP3⁻ and CD4-FOXP3⁺ T cells from Kitagawa et al (Nature Immunology 2017) and we do, indeed, observe H3K27 acetylation peak around Satb1-a region (See Figure 2B). Therefore, this strongly support our findings that Satb1-a (or Satb1-Eth2) functions as an enhancer for SATB1 in CD4 T cells.

5. The resolution of the Fig 3B Hi-C heatmap is too low, and the black line is too thick, obscuring the details of the SATB1 locus in the heatmap. Suggest a better presentation of the Hi-C data. In addition, the heat map of Hi-C could not provide the specific interaction between Satb1-a and the SATB1 promoter. It is recommended to perform a 4C assay or similar techniques to confirm the interaction between the enhancer and the promoter.

We apologise for poorly presenting the Hi-C data and thank Reviewer #3 for these suggestions. We have now added one additional Hi-C data of thymocytes and have provided Hi-C figures with increased quality/resolution (See Figure 3B). We have also provided 3C experimental data to show the interaction of Satb1-a enhancer with the Satb1 promoter.

6. The mice information in this manuscript is confusing. What is the difference between SATB1 Venus-Δa, SATB1 ΔEth2, and SATB1 Venus-ΔEth2? The authors should state this clearly in the manuscript. Since the CRISPR-Cas9 system excision is not perfect, there will be some variation in deletion regions. What is the difference between the deletion sequences in these mice? The authors should provide sequencing validation results for each mouse line.

We thank Reviewer #3 for this comment and apologise for the confusion. We modified the text to increase clarity.

Satb1-Δa and *Satb1-ΔEth2* are the same; we just changed the name from *Δa* to *ΔEth2* after confirming its role as an enhancer for Th2 cells (Eth2).

Likewise, *Satb1 Venus-Δa* and *Satb1 Venus-ΔEth2* are the same, but they both express SATB1-Venus fusion protein.

We provided sequences data for *Satb1 Venus-Δa* and *Satb1 Venus-Δb* mice in Fig EV2 in the original manuscript and added information of *Satb1-ΔEth2* mice on the B6 background.

Minor:

1. *Cd4-cre* (Line114, Page5) or *CD4-cre* (Figure EV1E)? Need to be consistent.
2. Figure 1C, 1D, 1E, and 1G have missing x-axis numbers.
3. Fig1F the conditions for FACS plots are not mentioned. The same issues are applicable to all FACS plots.
4. Page 7 Line 169: Figure "S2A" should be "EV2A".
5. Fig 4A right panel "*Satb1 ΔEth2ΔEth2*" should be "*Satb1 ΔEth2/ΔEth2*". The same issues are also applicable to Fig EV3A.
6. Page 12 "*Satb1 Δeth2/Δeth2*" should be "*Satb1 ΔEth2/ΔEth2*".

We thank Reviewer #3 for pointing out these minor issues of our manuscript and have attempted to correct all the above.

May 5, 2023

RE: Life Science Alliance Manuscript #LSA-2023-01897-TR

Dr. Ichiro Taniuchi
RIKEN Center for Integrative Medical Sciences
RIKEN Center for Integrative Medical Sciences
1-7-22, Suehiro-cho, Turumi-ku
Turumi-ku
Yokohama, Kanagawa 230-0045
Japan

Dear Dr. Taniuchi,

Thank you for submitting your revised manuscript entitled "Identification of a novel enhancer essential for Satb1 expression in TH2 cells and activated ILC2s.". We would be happy to publish your paper in Life Science Alliance pending final revisions necessary to meet our formatting guidelines.

- please address the final reviewers'2 and 3 points
- please upload your main and supplementary figures as single files
- please add a category for your manuscript to our system
- please add the Twitter handle of your host institute/organization as well as your own or/and one of the authors in our system
- please use the [10 author names, et al.] format in your references (i.e. limit the author names to the first 10)

A. FINAL FILES:

B. MANUSCRIPT ORGANIZATION AND FORMATTING:

Sincerely,

Reviewer #2 (Comments to the Authors (Required)):

The authors identified a new enhancer that promotes Satb1 expression in Th2 cells. First the authors characterized the expression of Satb1 in thymocytes and spleen cells using a satb1-venus fusion reporter. The authors correlate the expression of Venus with mRNA data and WB against Satb1. Secondly, the authors identified an enhancer that is necessary for the full expression of Satb1. The reduction of Satb1 expression is associated with an increase of IL5 in Th2 cells. Although the role of this enhancer in vivo is not described, the enhancer has a role in satb1 expression in ILC2 and Th2 cells. The authors have enriched the manuscript with data following the comments of the referees.

Minors concerns :

Figure 1B : It might be better to say that there is an increase in the number of cells expressing Satb1 rather than that Satb1 expression is increased from DN to DP.

Page 9 line 223 : « Satb1-a is essential ». May be say that the enhancer allow to maintain Satb1 expression at high level.

In general manner, it is necessary to clarify what is meant by 3 independent experiences. It would be nice to say in the materials and methods, how many mice are done per group and per independent experiments.

Reviewer #3 (Comments to the Authors (Required)):

In this manuscript, the authors use the constructed SATB1-Venus reporter mouse model to study the regulation mechanism of SATB1. The authors identified an enhancer regulating SATB1 in CD4+ Th2 cells, which can maintain the expression of SATB1 in CD4 Th2 cells and ILC2s. However, the authors did not observe the functional relevancy of the decrease of SATB1 expression caused by the loss of this enhancer on T cell function. Experiments in this study were well conducted, the conclusion is reliable, and it is valuable for understanding the transcriptional regulation of SATB1. All issues arising in the previous version have been resolved in this revision.

A small error in Fig. 2A. It should be 300kb instead of 300b.

Reviewer #2 (Comments to the Authors (Required)):

The authors identified a new enhancer that promotes Satb1 expression in Th2 cells. First the authors characterized the expression of Satb1 in thymocytes and spleen cells using a satb1-venus fusion reporter. The authors correlate the expression of Venus with mRNA data and WB against Satb1. Secondly, the authors identified an enhancer that is necessary for the full expression of Satb1. The reduction of Satb1 expression is associated with an increase of IL5 in Th2 cells. Although the role of this enhancer in vivo is not described, the enhancer has a role in satb1 expression in ILC2 and Th2 cells. The authors have enriched the manuscript with data following the comments of the referees.

We are very grateful for Reviewer #2's positive evaluation on our revised manuscript.

Minors concerns :

Figure 1B : It might be better to say that there is an increase in the number of cells expressing Satb1 rather than that Satb1 expression is increased from DN to DP.

We appreciate Reviewer #2's comment and have corrected the text in pages 5-6, lines 123-125.

Page 9 line 223 : « Satb1-a is essential ». May be say that the enhancer allow to maintain Satb1 expression at high level.

We appreciate this comment and have modified the text as per Reviewer #2's suggestion.

In general manner, it is necessary to clarify what is meant by 3 independent experiences. It would be nice to say in the materials and methods, how many mice are done per group and per independent experiments.

We thank Reviewer #3 for this suggestion and have now realised that using "independent experiments" could come across confusing to our readers.

Therefore, we have changed this term to "biological experiments" and have now elaborated its definition (i.e. number of mice used per experiment) in our methods section (page 19, lines 462-464).

Reviewer #3 (Comments to the Authors (Required)):

In this manuscript, the authors use the constructed SATB1-Venus reporter mouse model to study the regulation mechanism of SATB1. The authors identified an enhancer regulating SATB1 in CD4+ Th2 cells, which can maintain the expression of SATB1 in CD4 Th2 cells and ILC2s. However, the authors did not observe the functional relevancy of the decrease of SATB1 expression caused by the loss of this enhancer on T cell function. Experiments in this study

were well conducted, the conclusion is reliable, and it is valuable for understanding the transcriptional regulation of SATB1. All issues arising in the previous version have been resolved in this revision.

We are very relieved for Reviewer #3's positive evaluation on our revised manuscript.

A small error in Fig. 2A. It should be 300kb instead of 300b.

We appreciate this mistake pointed out by Reviewer #3 and have now corrected Fig 2A.

May 8, 2023

RE: Life Science Alliance Manuscript #LSA-2023-01897-TRR

Dr. Ichiro Taniuchi
RIKEN Center for Integrative Medical Sciences
RIKEN Center for Integrative Medical Sciences
1-7-22, Suehiro-cho, Turumi-ku
Turumi-ku
Yokohama, Kanagawa 230-0045
Japan

Dear Dr. Taniuchi,

Thank you for submitting your Research Article entitled "Identification of a novel enhancer essential for Satb1 expression in TH2 cells and activated ILC2s.". It is a pleasure to let you know that your manuscript is now accepted for publication in Life Science Alliance. Congratulations on this interesting work.

DISTRIBUTION OF MATERIALS:

Again, congratulations on a very nice paper. I hope you found the review process to be constructive and are pleased with how the manuscript was handled editorially. We look forward to future exciting submissions from your lab.

Sincerely,
